# DPPIV$^+$ fibro-adipogenic progenitors form the niche of adult skeletal muscle self-renewing resident macrophages

Farshad Babaeijandaghi [1,3,4] ✉, Nasim Kajabadi[1,4], Reece Long [1], Lin Wei Tung[1], Chun Wai Cheung[1], Morten Ritso [1], Chih-Kai Chang[1], Ryan Cheng [1], Tiffany Huang [1], Elena Groppa[1], Jean X. Jiang [2] & Fabio M. V. Rossi [1] ✉

Adult tissue-resident macrophages (RMs) are either maintained by blood monocytes or through self-renewal. While the presence of a nurturing niche is likely crucial to support the survival and function of self-renewing RMs, evidence regarding its nature is limited. Here, we identify fibro-adipogenic progenitors (FAPs) as the main source of colony-stimulating factor 1 (CSF1) in resting skeletal muscle. Using parabiosis in combination with FAP-deficient transgenic mice (*Pdgfrα$^{CreERT2}$* × DTA) or mice lacking FAP-derived CSF1 (*Pdgfrα$^{CreERT2}$* × *Csf1$^{flox/null}$*), we show that local CSF1 from FAPs is required for the survival of both TIM4$^-$ monocyte-derived and TIM4$^+$ self-renewing RMs in adult skeletal muscle. The spatial distribution and number of TIM4$^+$ RMs coincide with those of dipeptidyl peptidase IV (DPPIV)$^+$ FAPs, suggesting their role as CSF1-producing niche cells for self-renewing RMs. This finding identifies opportunities to precisely manipulate the function of self-renewing RMs in situ to further unravel their role in health and disease.

Tissue-resident macrophages (RMs) are a heterogenous population of immune cells found in almost all anatomical locations. RMs have well-defined tissue-specific homeostatic functions such as red blood cell disposal by splenic red pulp macrophages, bone resorption by osteoclasts, or synaptic remodeling by microglia[1–4]. In addition, further evidence has emerged regarding the important role of RMs in pathological conditions. For example, impaired clearance of autoantigens present on apoptotic cells by RMs has been proposed to drive the development of autoimmune reactions[5,6] and the expansion of RMs has been linked to the impairment of pancreatic β-cell function in obesity[7].

Skeletal muscle plays a critical role in metabolism and whole-body homeostasis[8]. Furthermore, it has a unique capacity to fully recover following injury, making it the subject of many studies on tissue regeneration[9–11]. Evidence is emerging regarding the important role of skeletal muscle RMs in muscle metabolism[12], growth[13], and aging[14]. In skeletal muscle, two main populations of RMs reside at steady state: a population of TIM4$^-$ macrophages derived from blood monocytes, and a population of TIM4$^+$ macrophages that self-renew (self-renewing resident macrophages, SRRMs)[12].

A niche, originally defined for self-renewing stem cells, is considered as an area of tissue composed of cells, matrix, and physical and molecular signals that provides a specific supportive microenvironment required for stem cell maintenance and function[15]. Similar to stem cells, SRRMs are likely located in a particular niche that governs their fate, phenotype and function[16]. However, while the niches of adult stem cells have been identified and characterized in multiple tissues[17–19], the existence of a niche for SRRMs has not yet been confirmed and little is known about its cellular and molecular components.

[1]Biomedical Research Centre, University of British Columbia, Vancouver BC V6T1Z3 BC, Canada. [2]Department of Biochemistry and Structural Biology, University of Texas Health Science Center, San Antonio TX 78229 TX, USA. [3]Present address: Altos Labs Inc, San Diego, CA, USA. [4]These authors contributed equally: Farshad Babaeijandaghi, Nasim Kajabadi. ✉e-mail: fbabaeijandaghi@altoslabs.com; Fabio@brc.UBC.ca

Fibro-Adipogenic Progenitors (FAPs) constitute a subset of mesenchymal cells residing in the interstitium of various tissues, including skeletal muscle. They possess a distinctive capacity to differentiate into fibrogenic and adipogenic lineages both in vivo and in vitro[20]. These cells can be identified within skeletal muscle by their expression of PDGFRα or as CD45-CD31-Sca-1+ cells[21,22]. In response to muscle injury, these cells become activated and undergo proliferation. While they don't directly contribute to myogenesis, they create a temporary environment for satellite cell activation and differentiation[21]. Upon activation, FAPs generate numerous cytokines, implying their potential involvement in modulating the inflammatory response. Beyond regeneration, a growing body of evidence underscores the pivotal role of FAPs in regulation of muscle homeostasis in both health and disease[23–26]. Here, we identify a subpopulation of FAPs that express dipeptidyl peptidase IV (DPPIV) as a key component of the niche for SRRMs in adult skeletal muscle.

## Results and discussion

### Colony-stimulating factor 1 (CSF1) from FAPs is required for the survival of monocyte-derived and self-renewing muscle RMs

It is well-known that CSF1 is required for the development and maintenance of RMs. The lack of CSF1 in op/op mice is associated with a reduced number of RMs in different tissues[27]. CSF1 from *Pdgfra*+ fibroblasts has recently been shown to be required for embryonic development of Kupfer cells in the liver[28,29]. The role of CSF1 is not limited to development but continues in adult tissues. CSF1R signaling is crucial for the survival of RMs in different adult tissues including skeletal muscle[12,30]. However, the source of CSF1 in adult skeletal muscle is not known. While circulating CSF1 has been proposed to nurture RMs in some tissues, CSF1 is provided by local resident cells in most tissues[31]. Therefore, in order to look for the potential existence of a nurturing niche for SRRMs in the skeletal muscle, we initially assessed the local cellular source of CSF1.

As we showed recently[12], skeletal muscle is composed of three main myelomonocytic populations (fig. S1a): a population of F4/80+TIM4+CCR2- resident macrophages (TIM4+ RMs) that self-renew (self-renewing resident macrophages, SRRMs), a population of F4/80+TIM4-CCR2-/+ resident macrophages (TIM4- RMs) that are in exchange with blood monocytes at steady state, and a population of F4/80LowCD11c+MHCII+CCR2+ cells likely representing dendritic cells (DCs). In contrast to the TIM4+ and TIM4- populations, the DC-like population has lower expression of *Csf1r* and is not dependent on CSF1R signaling to survive[12]. A similar population of self-renewing TIM4+ macrophages has also been shown to exist across different organs[32]. Different resident cell types have been proposed to be the local source of CSF1, including blood endothelial cells in the liver sinusoids[33], neurons in the gut[34], stellate cells in the liver[33], stromal fibroblasts and osteoblasts in the bone marrow[35], epithelial cells in the kidney proximal convoluted tubules[35], lymphatic endothelial cells in lymph nodes[36], reticular fibroblasts in the spleen[37], and osteoblastic cells in the bone[38]. To elucidate which cells are providing CSF1 in adult skeletal muscle, we initially evaluated the expression of *Csf1* across various muscle-resident cell types using two publicly available single-cell RNA sequencing datasets. In the Tabula Muris dataset[39], *Csf1* is expressed by muscle-resident mesenchymal stromal cells, including fibro-adipogenic progenitors (FAPs), endothelial cells, and tenocytes (Fig. S1b). On the other hand, in the dataset generated by De Micheli AJ et al.[40], *Csf1* expression is restricted to FAPs in skeletal muscle at steady state (Fig. S1c). Taking into account the limitations inherent in single-cell RNA sequencing such as potential biases in transcript coverage and the challenges posed by low sequencing depth, we further purified the main resident cell types including CD31+ endothelial cells, CD45+ hematopoietic cells, CD31-CD45-Sca1-CD106+ skeletal muscle precursor cells (MuSCs)[41], and CD31-CD45-Sca1+ FAPs[42], from undamaged skeletal muscle using fluorescence-activated cell sorting (FACS; Fig. 1a) to

assess *Csf1* expression using droplet digital PCR. We found FAPs to be the only cell population expressing high levels of *Csf1* at steady state (Fig. 1b). Using *Pdgfra*CreERT2 × DTA mice, we successfully achieved the depletion of 80% of FAPs within one month following initiation of tamoxifen induction (Fig. 1c). Depletion of FAPs resulted in a significant reduction in the amount of CSF1 protein in skeletal muscle, further supporting that FAPs are a main source of CSF1 production in skeletal muscle at steady state (Fig. 1d). Within two weeks of FAP depletion (one month after tamoxifen initiation), there was a comparable reduction in the number of both TIM4+ SRRMs and blood-derived TIM4- RMs, suggesting that FAPs are required for their survival (Fig. 1e). To determine the role of FAP-derived CSF1 in the maintenance of RMs, we generated a mouse strain in which one allele of *Csf1* has been deleted and the other allele is flanked by LoxP sites (*Csf1*flox/null). In order to specifically target the deletion of *Csf1* from FAPs, we crossbred *Csf1*flox/null mice with *Pdgfra*CreERT2 mice, generating a mouse strain herein referred to as the *Csf1*flox mouse strain. This approach enabled us to delete both alleles of *Csf1* in approximately 72% of FAPs after tamoxifen induction (Fig. S1d). One month after initiation of tamoxifen induction, we observed no notable alterations in the number of TIM4- RMs and TIM4+ RMs in *Csf1*flox mice. Within 2–3 months, however, we observed a significant reduction in the number of TIM4+ SRRMs (Fig. 1f), confirming that CSF1 from FAPs is required for the survival/self-renewal of this population. There was also a reduction, albeit to a lesser extent, in the number of TIM4- RMs (Fig. 1f). Pharmacological inhibition of CSF1R signaling[12] or depletion of FAPs (Fig. 1e) led to a swift reduction in RMs. However, RMs exhibited prolonged survival after the depletion of CSF1 from FAPs. This might be attributed, in part, to the relatively lower recombination efficiency of the *Csf1*flox system as well as using mice with a null allele of *Csf1* as controls. It may also indicate that FAPs supply additional physical and/or molecular signals capable of temporarily compensating for the lack of CSF1, thereby supporting the survival of RMs. Using mice with a null allele of *Csf1* as controls could also potentially explain the relatively modest reduction in the number of RMs observed in this model.

As mentioned above, in contrast to TIM4+ RMs that self-renew at steady state, TIM4- RMs are replaced by blood monocytes. As PDGFRα+ cells exist in other tissues including the bone marrow[43], we asked whether the reduced numbers of muscle TIM4- macrophages in *Csf1*flox mice resulted from deletion of CSF1 from FAPs in the bone marrow. To this end, we measured CSF1 in the bone marrow of the induced FAP-deficient transgenic mice (*Pdgfra*CreERT2 × DTA) mice, which showed no difference compared to transgenic (Tg) control (Fig. S1e). Additionally, we surgically joined wild type (WT) B6.SJL-Ptprca Pepcb/BoyJ (B6 CD45.1) mice in parabiosis with tamoxifen-induced *Csf1*flox mice to assess whether WT bone marrow can rescue the number of TIM4- RMs of the *Csf1*flox partner. A blood sharing rate of 19.22% ± 2.8% (mean ± standard deviation) was achieved in the *Csf1*flox partner (Fig. S1f). The number of TIM4- RMs in the *Csf1*flox partner remained lower than the number of TIM4- RMs in the WT partner after two months of blood sharing (Fig. 1g) suggesting that their depletion resulted from the absence of local CSF1 rather than from their impaired generation in the bone marrow. In summary, our data suggest that local CSF1 from muscle resident FAPs are required for the survival of both monocyte-derived TIM4- and self-renewing TIM4+ RMs.

### Local CSF1 is required for restoration of depleted RMs

The number of RMs has been hypothesized to be controlled by the local availability of CSF1. According to this model[31], following macrophage death, the amount of available CSF1 increases either passively due to the absence of CSF1 consumption or through active production of CSF1 by the niche cells. The increased availability of CSF1 induces macrophage proliferation and consumption of CSF1 by newly generated macrophages restores the initial equilibrium. To assess whether there is any change in CSF1 protein levels following RM depletion, we

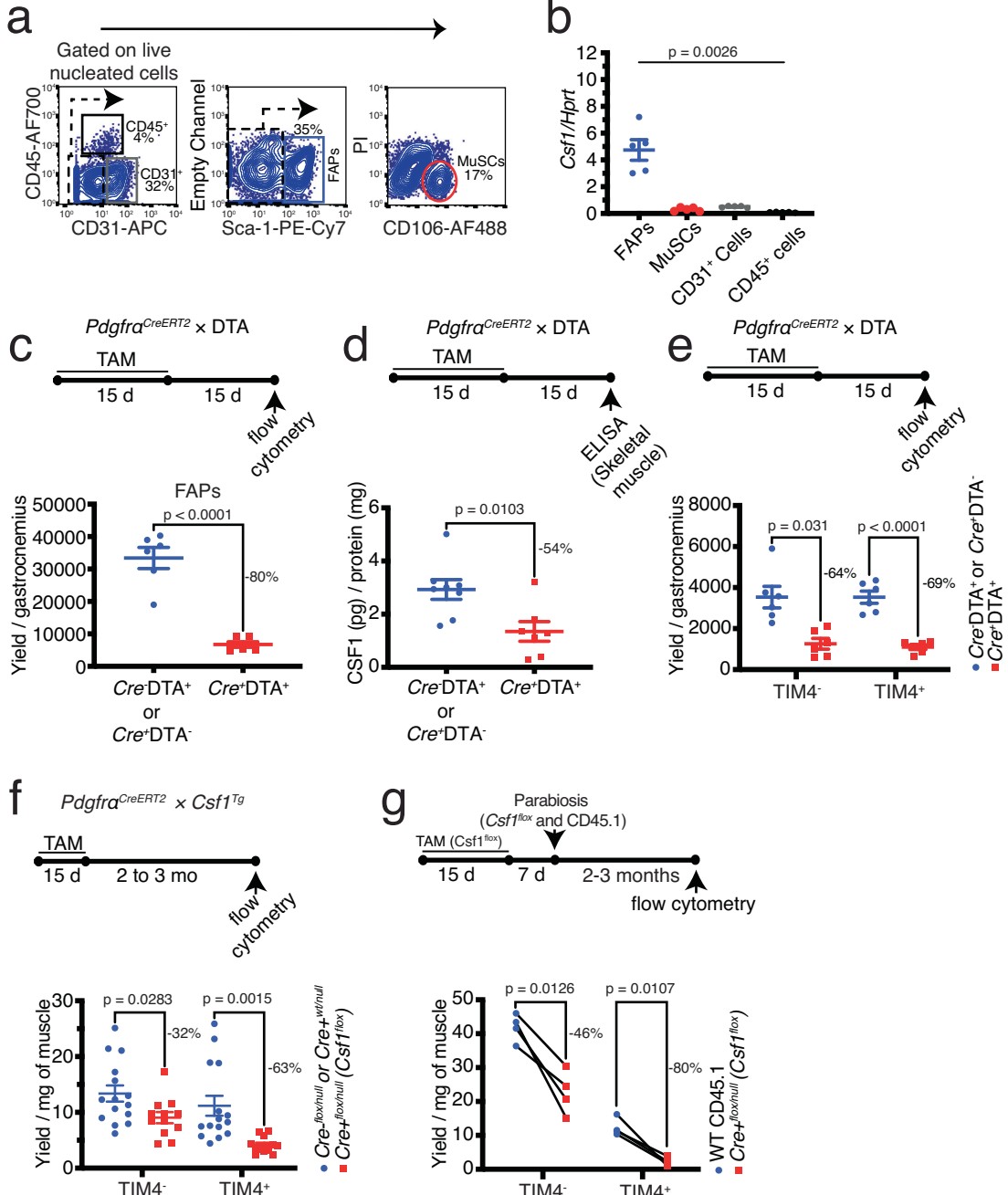

**Fig. 1 | Local CSF1 from FAPs is required for the survival of muscle RMs. a** Gating strategy to identify different muscle resident cell populations. **b** Expression of Csf1 in different muscle resident cell populations at steady state assessed by droplet digital PCR (n = 5 mice pooled from 2 experiments, Brown-Forsythe and Welch ANOVA tests). **c** The number of FAPs detected by flow cytometry in skeletal muscle from tamoxifen (TAM)-induced PdgfraCreERT2 × DTA mice compared to TAM-induced transgene (Tg) controls (each dot represents one mouse, data were pooled from 2 experiments, unpaired *t*-test). Because there is a minor decrease in muscle mass after depleting FAPs, their numbers are presented per muscle (gastrocnemius) and not per milligram of tissue. **d** The amount of CSF1 detected by ELISA in quadriceps muscle of TAM-induced PdgfraCreERT2 × DTA mice compared to Tg

control (each dot represents one mouse, data were pooled from ≥3 experiments, unpaired *t*-test). **e** The number of TIM4- and TIM4+ RMs detected by flow cytometry in skeletal muscle from TAM-induced PdgfraCreERT2 × DTA mice compared to Tg control (each dot represents one mouse, data were pooled from 2 experiments, unpaired *t*-test). **f** The number of TIM4- and TIM4+ RMs detected by flow cytometry in skeletal muscle from TAM-induced PdgfraCreERT2 × Csf1flox/null mice and Tg controls (each dot represents one mouse, data were pooled from ≥3 experiments, unpaired *t*-test for TIM4- and unpaired *t* test with Welch's correction for TIM4+). **g** Replacement of depleted muscle RMs in TAM-induced Pdgfra-CreERT2 × Csf1flox/null mice by bone marrow blood progenitors from the wild type (WT) parabiotic partner (n = 4 parabiotic pairs from ≥3 experiments, paired *t*-test).

measured CSF1 in skeletal muscle at different time points after CSF1R inhibition and withdrawal (inh/wd) using ELISA. There was a significant increase in CSF1 concentration in skeletal muscle following CSF1R inhibition, which returned to baseline levels within 7 days of withdrawing the inhibitor (Fig. 2a). As circulating CSF1 has been proposed

to regulate RM homeostasis in some tissues, we further measured the amount of CSF1 in the serum. We did not detect any changes in serum CSF1 concentration following CSF1R inh/wd (Fig. 2b), suggesting that the level of CSF1 in skeletal muscle is controlled locally. Therefore, we further investigated whether additional local sources of CSF1 are

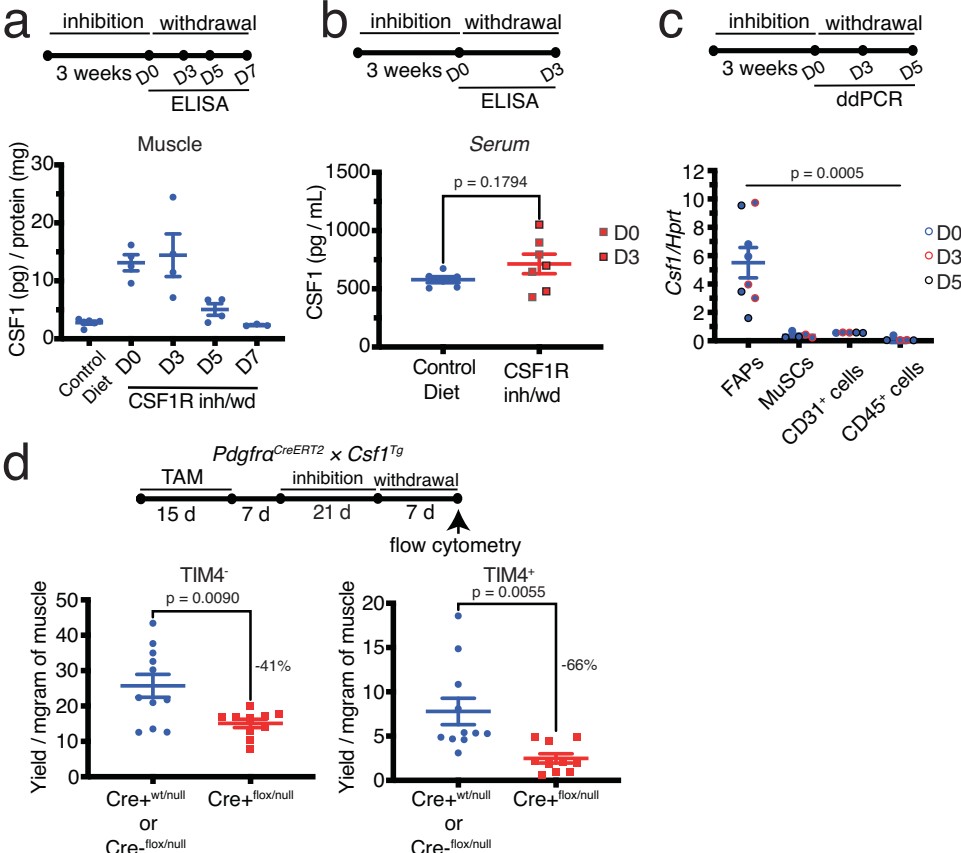

**Fig. 2 | Local CSF1 restores depleted RMs. a** The amount of CSF1 detected by ELISA in skeletal muscle at different time points after CSF1R inh/wd (each dot represents one mouse, data were pooled from ≥3 experiments). **b** The amount of CSF1 detected by ELISA in the blood within three days after withdrawal of CSF1R inhibition or the control diet (each dot represents one mouse, data were pooled from 2 experiments, unpaired *t*-test). **c** Expression of Csf1 in different muscle resident cell populations at different time points after CSF1R inh/wd assessed by droplet digital PCR (ddPCR, *n* = 8 mice for FAPs, *n* = 5 mice for the other populations, data were pooled from ≥3 experiments, Brown-Forsythe and Welch ANOVA tests). **d** The number of TIM4- and TIM4+ RMs detected by flow cytometry in skeletal muscle from tamoxifen (TAM)-induced PdgfrαCreERT2 × Csf1flox/null or transgene (Tg) controls following CSF1R inh/wd (*n* = 11 for control and *n* = 10 for CSF1R inh/wd, data were pooled from ≥3 experiments, unpaired *t* test with Welch's correction).

recruited to replenish RMs. Following CSF1R inh/wd, *Csf1* expression was still limited to FAPs among the cell populations that we analyzed (Fig. 2c). As depicted in Fig. 1f, a decreased number of RMs is evident upon targeted depletion of CSF1 from FAPs in steady-state muscle (32% and 63% reduction for TIM4- and TIM4+ RMs, respectively). This reduction becomes more pronounced after CSF1R inh/wd, with a 41% and 66% reduction for TIM4- and TIM4+ RMs, respectively (Fig. 2d). A lack of correlation between *Csf1* mRNA and protein levels has been suggested by different groups[44,45]. On the other hand, measuring the amounts of secreted factors from primary cells poses a challenge due to their extremely low concentrations. While direct analysis of CSF1 protein content in FAPs after RM depletion was not feasible, our observations, combined with the absence of noteworthy changes in the number of FAPs following inhibition of CSF1R (Fig. S2a), suggest that FAPs regulate local CSF1 production to replenish adult skeletal muscle RMs following their depletion.

**DPPIV+ FAPs contribute to the niche of skeletal muscle SRRMs**
Following drug withdrawal, the number of blood-derived TIM4- cells overshoots their pretreatment abundance three to five folds (Fig. 3a, and[12]), while the number of TIM4+ SRRMs is tightly controlled and remained close to pre-treatment levels through an intrinsic or extrinsic mechanism. At steady state, TIM4+ macrophages repopulate themselves through self-renewal with small contribution from blood monocytes[12]. Their intrinsic self-renewal capacity could also

potentially play a role in their restoration following withdrawal of pharmacological inhibition, as well as in the regulation of their numbers. Alternatively, their replacement following pharmacological depletion could rely on contributions from circulating precursors/ TIM4- macrophages. To assess whether blood-derived, TIM4- macrophages are capable of directly contributing to TIM4+ SRRMs, we initially treated C57BL/6 J mice with a CSF1R inhibitor to ablate TIM4+ SRRMs. Then, we withdrew the inhibition 24 h before injecting (transplanting) sorted skeletal muscle GFP+TIM4- RMs into the Tibialis Anterior (TA) muscle. Six days after transplantation, we analyzed whether any of the injected GFP+TIM4- cells were able to differentiate into TIM4+ cells. Macrophages exhibit poor engraftment following intramuscular injection, and we were able to only detect a small number of engrafted cells out of thousands of cells that were administered in one of our attempts. This preliminary data, however, demonstrated an increase in TIM4 expression within a portion of the recovered GFP+TIM4- cells (Fig. S3a). This implies the potential for TIM4- cells to differentiate into TIM4+ cells under these circumstances. To further confirm this finding and assess the extent of contribution from bloodborne monocytes to TIM4+ cells following CSF1R inh/wd, we surgically created parabiotic pairs of C57BL/6 J (CD45.2) and B6.SJL-Ptprca Pepcb/BoyJ (B6 CD45.1) mice or C57BL/6 J and mice ubiquitously expressing green fluorescent protein (GFP+). Once blood sharing was established, we treated the animals with the CSF1R inhibitor for three weeks. One to two weeks after drug withdrawal, we

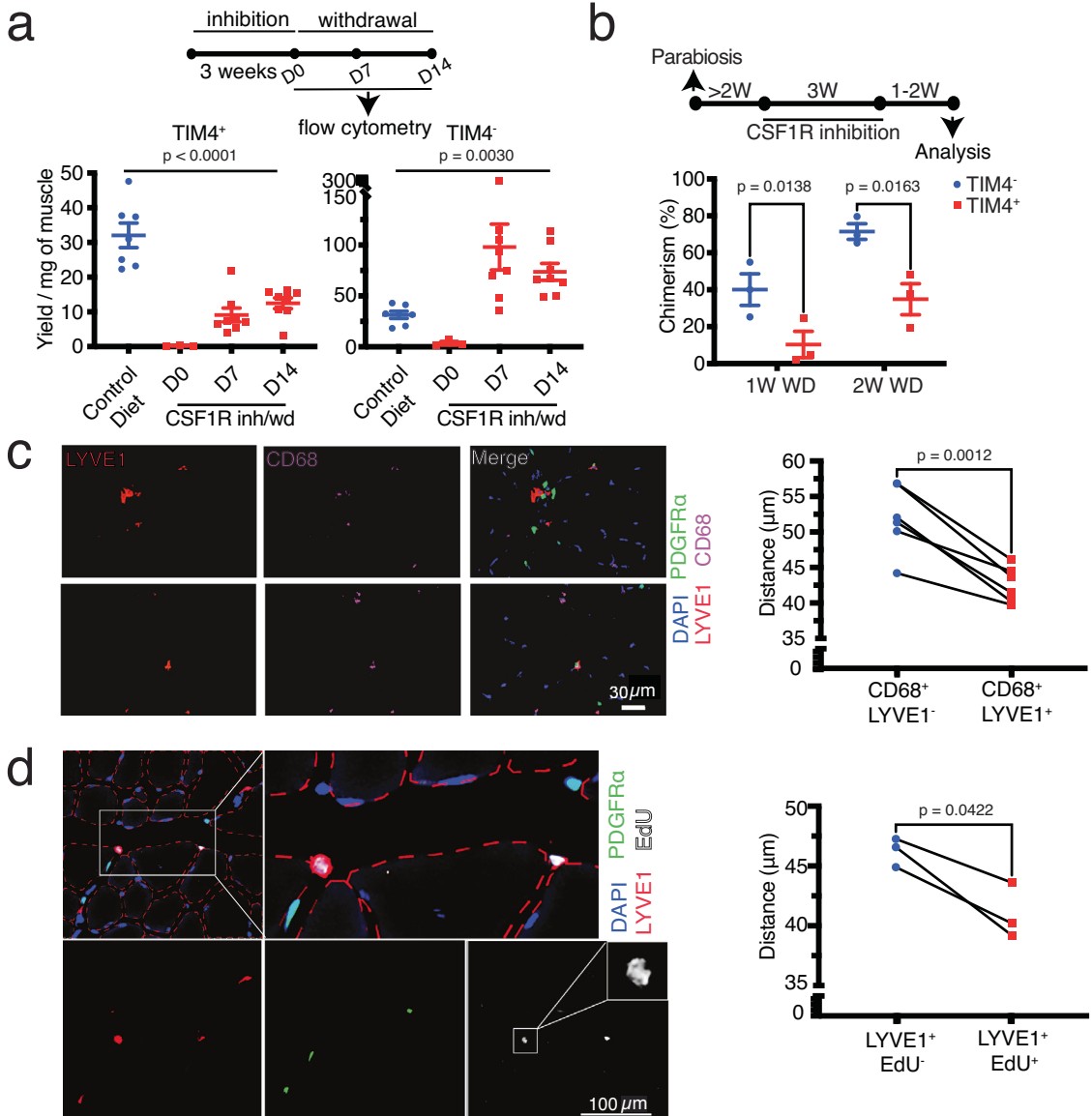

**Fig. 3 | SRRMs are located close to FAPs. a** The number of TIM4- and TIM4+ RMs detected by flow cytometry in skeletal muscle following CSF1R inh/wd (*n* = 7 mice for the control group, *n* = 3, 8, and 8 for the CSF1R inh/wd group on D0, D7, and D14, respectively; data were pooled from ≥3 experiments, ordinary one-way ANOVA). **b** Replacement rate of muscle RMs by blood progenitors following pharmacological depletion: The replacement rate of RMs was normalized to the percentage of chimerism in mononuclear myelomonocytic cells in the blood (*n* = 3 female parabiotic pairs per time point, data were pooled from ≥3 experiments, paired *t* test). **c** Immunofluorescent staining of Tibialis Anterior (TA) muscle sections and quantification of the distance between LYVE1+ SRRMs or LYVE1- RMs and

PDGFRα-EGFP+ FAPs (*n* = 6 mice, pooled from 2 experiments; paired *t*-test). On average, we manually measured distances of approximately 575 CD68 + LYVE1- and 450 CD68 + LYVE1+ RMs from their closest FAPs in each mouse.
**d** Immunofluorescent staining of TA sections and quantification of the distance between LYVE1+EdU+ or LYVE1+EdU- cells and PDGFRα-EGFP+ FAPs (*n* = 3 mice pooled from 2 experiments; paired *t*-test). On average, we manually measured distances of approximately 175 LYVE1+EdU- and 25 LYVE1+ EdU+ cells from their closest FAPs in each mouse. The figure displays two LYVE1+ EdU+ cells situated in proximity to FAPs. The cell on the left exhibits a dividing nucleus, as illustrated in greater detail in the inset.

analyzed the contribution of blood progenitors to the repopulation of RMs (Fig. S3b). As expected, TIM4- macrophages were mostly repopulated by blood progenitors (Fig. 3b). In contrast, less than half of the TIM4+ population originated from the bloodstream, indicating that some degree of local self-renewal is involved in their restoration (Fig. 3b). Conversely, the noticeable number of blood-derived TIM4+ macrophages following CSF1R inh/wd also suggests the involvement of an extrinsic factor in governing the number of TIM4+ macrophages.

As FAPs are the main source of CSF1 in skeletal muscle, it is possible that they could also be involved in controlling the number of TIM4+ SRRMs. To this end, we hypothesized that FAPs may locate in close proximity to TIM4+ macrophages to "sense" their numbers. To

test this hypothesis, we evaluated the location of TIM4+ SRRMs relative to FAPs in muscle in PDGFRα[EGFP] mice. As we could not find an antibody for TIM4 that works well on skeletal muscle sections, and since in contrast to TIM4- RMs, all TIM4+ SRRMs are also positive for LYVE1 (lymphatic vessel endothelial receptor 1) at steady state[12], we used this marker to identify TIM4+ SRRMs in histological sections. Although lymphatic vessels also express LYVE1, they can be easily identified based on their lack of CD68 expression (a pan marker for macrophages) or vascular morphology. We found that CD68+LYVE1+ macrophages are situated in closer proximity of FAPs compared to CD68+LYVE1- cells in homeostatic skeletal muscle (Fig. 3c). On the other hand, both populations were located at a comparable distance

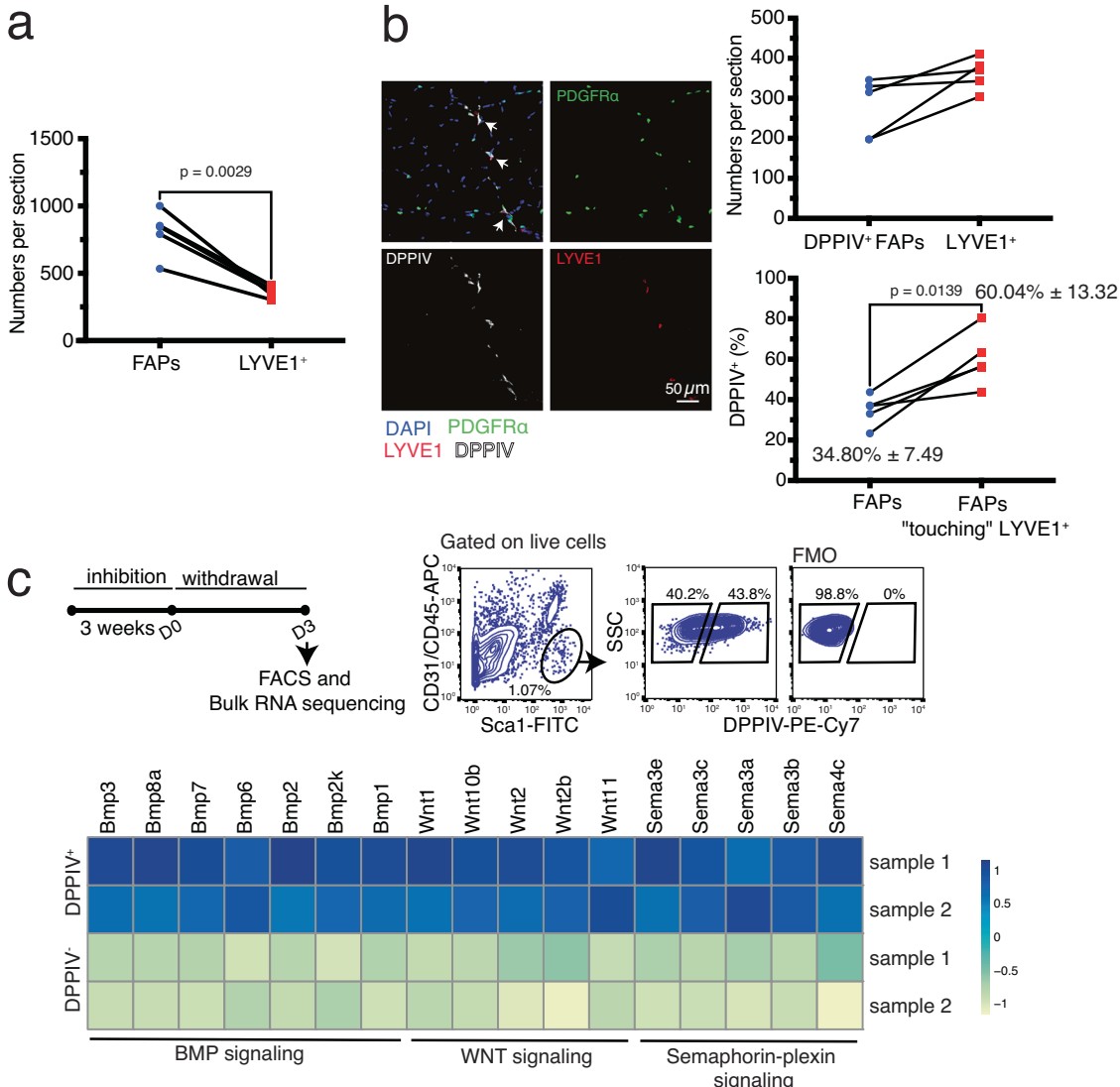

**Fig. 4 | DPPIV+ FAPs form the niche of skeletal muscle SRRMs. a** The number of LYVE1+ SRRMs and FAPs (PDGFRα-EGFP+) in TA sections (*n* = 4 mice pooled from two experiments, paired *t*-test). **b** Immunofluorescent staining of TA sections and quantification of the number of LYVE1+ SRRMs and DPPIV + PDGFRα-EGFP+ FAPs (top) and the percentage of DPPIV+ cells among all PDGFRα-EGFP+ FAPs or among PDGFRα-EGFP+ FAPs with a SRRM in their close vicinity (*n* = 5 mice pooled from 2 experiments, paired *t*-test, bottom). On average, approximately 800 FAPs per mouse were included for quantification. Among them, an average of about 100 FAPs had SRRMs in close vicinity. **c** Experimental design and gating strategy to sort DPPIV- and DPPIV+ FAPs for bulk RNA sequencing (top) and heatmap showing selected differentially upregulated genes in DPPIV+ FAPs compared to DPPIV- FAPs categorized by different signaling pathways (bottom).

from endothelial cells, which constitutes another abundant population of muscle resident stromal cells (Fig. S3c). Skeletal muscle TIM4+ SRRMs are continuously proliferating at steady state[12]. We reasoned that proliferation of TIM4+ SRRMs should happen in proximity of their niche to receive the required proliferation-promoting signals. Therefore, we treated PDGFRα[EGFP] mice with 5-ethynyl-2′-deoxyuridine (EdU) in drinking water (0.5 mg/mL) for two weeks and then assessed the location of EdU+ SRRMs compared to FAPs. We found FAPs were located closer to EdU+ proliferating LYVE1+ cells than nonproliferating LYVE1+ cells (Fig. 3d), which further reinforces their potential role as niche cells.

The number of self-renewing stem cells in a tissue is often controlled by a limited number of niches[46,47]. Therefore, if FAPs form the niche of SRRMs, their numbers should be equal to or lower than the number of SRRMs. However, in skeletal muscle, the number of FAPs is higher than the number of TIM4+ SRRMs (Fig. 4a), suggesting that only a specific subpopulation of FAPs may form the niche of TIM4+ SRRMs. To address this hypothesis we initially compared the expression profile

between *Csf1* positive and negative FAPs using the single-cell RNA sequencing data generated from a previous study[48]. Among the differentially expressed genes, we found *Dpp4*, which encodes for the surface marker multifunctional protein dipeptidyl peptidase 4 (DPPIV, Fig. S4a), was highly expressed in *Csf1*-expressing FAPs. Interestingly, multiple studies have distinguished skeletal muscle FAPs into *Dpp4+* and *Dpp4-* subsets, both of which are functionally distinct[48,49]. Therefore, we further characterized DDPIV expression in different cell populations in skeletal muscle using flow cytometry. Consistent with the transcriptomic data, we found two populations of DPPIV- and DPPIV+ FAPs (Fig. S4b). In contrast, DPPIV expression was not found by MuSCs or endothelial cells (Fig. S4b). DPPIV was also expressed by a subpopulation of CD45+ cells. However, further analysis revealed that DPPIV is not expressed by SRRMs and among resident myelomonocytic cells, DPPIV expression is mainly limited to a subpopulation of CD11c+ dendritic-like cells (Fig. S4c). The absence of DPPIV expression on SRRMs (LYVE1+ macrophages in our flow cytometry analysis, Fig. S4c) allowed us to confidently study their location relative to DPPIV+

FAPs in muscle histology sections. We found that the number of DPPIV[+] FAPs was equal to or lower than the number of SRRMs (Fig. 4b). Although only 34.80% ± 7.49 (mean ± standard deviation) of FAPs express DPPIV in our histological analysis, the majority of FAPs (60.04% ± 13.32) that have an SRRM in their immediate vicinity express DPPIV (Fig. 4b). More importantly, our RNA sequencing data three days after withdrawal of CSF1R inhibition, a time at which the niche of SRRMs is empty and receptive, showed that DPPIV[+] FAPs expressed high levels of Bone Morphogenetic Proteins (BMP) ligands, Wnt ligands, and semaphorins compared to DPPIV[-] FAPs (Fig. 4c). BMP signaling plays a role in the differentiation of liver RMs (Kupffer cells) and mucosal Langerhans cells[33,50]. Additionally, spatial distribution of Wnt molecules and Wnt inhibitors has been proposed to be involved in the differentiation of blood-derived monocytes into tissue macrophages[51]. Semaphorins are soluble proteins that were initially identified to be involved in guiding axonal migration during neuronal development. The biological roles of this sub-family of proteins are growing, and recent evidence suggested their involvement in migration and activation of different immune cells including neutrophils, dendritic cells, and macrophages[52]. While further studies are needed to evaluate the precise role of niche-derived signaling molecules on muscle RMs, our findings together suggest that DPPIV[+] FAPs are a part of the SRRM niche involved in their differentiation and controlling their numbers.

Three models have been proposed to explain how the number of RMs is controlled in a tissue. In the first model, as was explained earlier, the number of macrophages is controlled by the local availability of CSF1[31]. In the second model, known as the territory model, the number of RMs is solely controlled by physical niche availability and contact inhibition[16]. The third model, the revised version of the two previous models, suggests that the number of RMs is controlled by the number of CSF1-containing niches[31]. According to this model, CSF1 is only present in the niche either as a membrane-bound isoform or due to the fixation of the soluble isoform to the ECM. Macrophages would be attracted towards an empty niche by the trophic effect of CSF1. They would then occupy the niche, consume CSF1 and physically repel the others. Our data supports the presence of CSF1 producing niches (DPPIV[+] FAPs) for SRRMs in skeletal muscle. However, CSF1 production is not limited to the niche as both DPPIV[-] and DPPIV[+] FAPs express *Csf1* (Fig. S4d) and its overproduction results in overexpansion of blood-derived RMs. Therefore, while niche-derived CSF1 is indispensable for the maintenance of SRRMs, other niche-factors (physical or chemical) appear to be involved in controlling their numbers, in contrast to what has also been recently proposed based on in-vitro coculture experiments[28,29].

In summary, this study provides further evidence on the existence of a niche for self-maintained RMs and sheds light on its cellular composition in skeletal muscle. Our findings support the emerging concept of fibroblasts providing a supportive microenvironment for macrophages in different organs at steady state and in inflammatory conditions through the CSF1-CSF1R axis[53]. While different strategies exist to deplete RMs, tissue-specific targeting of RMs without influencing the surrounding tissue is challenging[31]. A better understanding of the niche of RMs offers novel opportunities to precisely manipulate their functions in situ to further unravel their role in health and disease.

## Methods

### Mice
Animals were housed in standard housing conditions in a specific pathogen free facility. All animal procedures were approved by the University of British Columbia Animal Care Committee under experimental protocol numbers A18-0314 and A19-0316. Adult mice of both sexes, aged between 7 and 24 weeks were used in this study. We used age- and sex-matched littermates randomly assigned to different experimental groups wherever possible. C57BL/6 J (#000664), B6

CD45.1 (#002014), ROSA-DTA (#009669)[54], and PDGFRα[EGF] (#007669)[55] mice were purchased from The Jackson Laboratory. C57BL/6 mice expressing GFP ubiquitously from a cytomegalovirus–β-actin hybrid promoter were a gift of I. L. Weissman (Stanford University)[56]. To generate *Csf1[null/flox]* mice, non-tissue-specific null alleles of *Csf1* were obtained through a cross between Tie2-Cre (The Jackson Laboratory, #008863)[57] with *Csf1[flox/flox]* mice[58] (a gift of S.A. Werner and J.X. Jiang, University of Texas Health Science Center at San Antoni). Tie2-Cre recombinase activity is detected in endothelial tissues, as well as reproductive tissues, leading to the germline deletion of the floxed allele. Subsequently, we selected *Csf1[null/flox]* mice and crossed them with *Pdgfra[CreERT2]* mice (The Jackson Laboratory, #032770)[59] to conditionally delete the Csf1 floxed allele in FAPs. CRE activation was achieved by feeding mice with Tamoxifen Diet (250 mg/kg, TD.130856, Envigo) for 15 days. PLX5622 hemifumarate (MedChemExpress, # HY-114153A, Fig. S2 and Fig. S5) or PLX73086 (Plexxikon Inc., for all other experiments) was used for CSF1R inhibition. They were given in chow (200 mg/kg for PLX73086 and 1200 ppm free base for PLX5562) for 3 weeks. Both compounds were efficient in depleting muscle RMs (Fig. 3a (PLX73086) and Fig. S5 (PLX5562)).

### Tissue preparation for flow cytometry and cell sorting
Mice were euthanized using either $CO_2$ inhalation or via intraperitoneal injection of 2,2,2-tribromoethyl alcohol (Aldrich T4840-2). To avoid confounding results, prior to tissue collection for flow cytometric analysis of RMs, mice were perfused with 15 mL of warm PBS to remove circulating cells from blood vessels. Following removing the fascia, skeletal muscle from both hindlimbs of the mice were then carefully dissected. A mixture of Collagenase D (Roche; 1.5 U ml-1; # 11088882001) and Dispase II (Roche; 2.4 U ml-1; # 04942078001), and 10 mM CaCl2 was added to the samples and was incubated at 37 °C for 60 min. During the incubation period, the muscle mixtures were vortexed initially and then once every 20 min, for a total of four times. Preparations were passed through a 40 μm cell strainer (Falcon™, # 352340) and washed with PBS. This filtration created a single suspension of cells which was then centrifuged for 7 minutes at 550 g. In preparation for flow cytometry, blocking buffer, composed of 10% goat serum (GeneTex, # GTX73249), flow cytometry staining (FACS) buffer (PBS, 2% FBS, 2 mM EDTA) and 1:200 Anti-Fcγ Receptor (AbLab, clone 24G2, # 21-0041-05) were added to each falcon tube containing the single cell suspensions. A cocktail of antibodies (Supplementary Table 1) was then added to each cell suspension in the appropriate dilutions. Stained cells were washed and analyzed using a LSRII (Becton Dickinson, FACSDiva™ V6.1.3 software) or CytoFLEX (Beckman Coulter, CytExpert v2.4 software). Cell sorting was performed using a FACSAria (FACSDiva™ V6.1.3 software) or Influx cell sorter (Becton Dickinson, FACS™ Software v1.0.0.650). Flow cytometry plots were generated using FlowJo Engine software v5.00000.

### Immunofluorescence histology staining
Frozen muscle sections were rehydrated with PBS and then soaked in blocking buffer (2% normal goat serum in PBST, where PBST is made up of PBS and 0.3% Triton X 100) for one hour. Each section was then soaked in a mixture of primary antibody in blocking buffer (2% normal goat serum or normal donkey serum in PBS) for one hour, before being washed with PBST. Next, incubation with proper secondary antibodies was performed one hour followed by washing and nuclear staining with DAPI. The slides were covered with a cover slip and Fluoromount G (Southern Biotech, # 0100-01) and imaged using either Nikon Eclipse (NIS Elements v5.11.03 software) or Zeiss LSM900 confocal (ZEN Blue v3.4.91.00000 software) microscopes. Semi-automatic analysis of images was performed using either the ImageJ (Version 2.0.0-rc-69/1.52p) or the Nikon NIS-Element v5.11.03 software. A list of the antibodies used is summarized in Supplementary Table 1.

## Detection of EdU incorporation on histology sections

Initially, frozen sections were fixed by 4% PFA. Following PBS washing, EdU staining was performed using Click-iT™ Plus EdU Cell Proliferation Kit for Imaging, Alexa Fluor™ 647 (Invitrogen, # C10640) accordance to the manual. Finally, staining for surface markers was performed as mentioned above.

## Droplet digital PCR

To assess the expression of *Csf1* in different cell populations, up to 30,000 cells were sorted directly into RLT buffer. After RNA isolation (RNeasy micro kit, Qiagen, #74034) and reverse transcription (Superscript III Reverse Transcriptase, Life Technologies), cDNA was diluted with TE buffer and 9.9 µl was used in a reaction mix containing Droplet Digital PCR Supermix (Bio-Rad), 1× TaqMan assay (Mm00432686_m1) and H2O. Droplets were generated with a QX100 droplet generator (Bio-Rad), after mixing 20 µl of reaction mix and 70 µl of droplet generator oil (Bio-Rad). The emulsified samples were loaded onto 96-well plates and endpoint PCRs were performed in C1000 Touch thermal cycler (Bio-Rad) at the following cycling conditions: 95 °C for 10 min, followed by 45 cycles at 94 °C for 30 s and 60 °C for 1 min, followed by 98 °C for 10 min. The droplets from each sample were read through the QX100 droplet reader (Bio-Rad). Resulting PCR-positive and PCR-negative droplets were counted using QuantaSoft software v1.7.4.0917 (Bio-Rad). Expression levels were normalized to hypoxanthine-guanine phosphoribosyltransferase [*Hprt*, Taqman assay (Mm00446968_m1)]. The efficiency of the Cre-Lox system to specifically delete *Csf1* gene, DNA was extracted from double sorted FAPs using QIAamp DNA micro kit (# 56304). Extracted DNA was used in a reaction mix containing Droplet Digital PCR Supermix (Bio-Rad), 1× TaqMan assay (Mm00547704_cn) and H20 for downstream droplet generation and PCR amplification as mentioned above. *Csf1* copy numbers were normalized to Catenin Beta 1 [*Cnntb1* (TaqMan assay (Mm00274200_cn)].

## CSF1 detection

To detect CSF1 in skeletal muscle, quadriceps muscle was cut into small pieces and submerged in 300–350 µL of lysis buffer (20 mM Tris pH 7.8, 137 mM NaCL, 2.7 mM MgCL2, 1% Triton x-100, 10% glycerol, 1 mM EDTA, 1 mM DTT, 1% Protease inhibitor). The samples were homogenized using metal beads with TissueLyser II (Qiagen) for 12 min at 20 Hz followed by 15 cycles of medium intensity sonication in Bioruptor® Plus sonication device at 4 degrees. Following homogenization, the samples were centrifuged at 10,000 g for 10 min at 4 degrees. 150 µL of supernatant was used for downstream applications. To detect circulating CSF1, blood samples were centrifuged at 10,000 g for 10 min at 4 degrees. 50 µL of the serum was directly used for downstream applications. To detect CSF1 in the bone marrow, the bone marrow of a Tibia and a femur was flushed out using 250 µL of the lysis buffer. Following vigorous vortexing, the samples were centrifuged at 10,000 g for 10 min at 4 degrees and the supernatant was used for downstream applications. Initially, the amount of protein in each sample was quantified using Pierce™ BCA Protein Assay Kit (Thermo Scientific, #23228). Finally, the concentration of CSF1 in 150 µL of muscle lysate, 50 µL of serum, or 50 µL of bone marrow lysate (diluted in 100 µL, 250 µL, or 200 µL of Assay Diluent RD1-38, respectively) was determined using the Mouse M-CSF Quantikine ELISA Kit (R&D Systems, #MMC00), following the manufacturer's instructions. Signal detection and quantification were carried out using Tecan infinite M200 Pro ELISA-Reader and Magelan pro v7.3 software.

## Bulk RNA sequencing and analysis

RNeasy Micro Kit (Qiagen, # 74004) was used to isolate RNA from sorted cells. Sample quality was assessed using the Agilent 2100 Bioanalyzer. RNA samples with an RNA Integrity Number >8 were prepared in accordance with the protocol for the TruSeq Stranded mRNA library kit (Illumina) on the Illumina Neoprep automated microfluidic library prep instrument as was previously described[11]. Paired-end sequencing was performed on the Illumina NextSeq 2000 using the P2 reagents (100 cycle; Illumina). Illumina base call files were de-multiplexed by bcl2fastq2 (v2.20) on Basespace. Demultiplexed read sequences were then aligned to the mm10 genome reference using STAR aligner. The number of aligned reads to each annotated gene was tallied with RnaReadCounter to generate read-count matrices for all samples, which were used as inputs for downstream analyses. Bowtie, STAR, and RnaReadCounter are tools built under RNA-Seq Alignment (v1.1.1). Downstream analyses of read-count data were performed in R (v3.6.2). Genes with less than 2 counts per million (CPMs) in at least two samples were filtered out. Filtered counts were processed and analyzed using DESeq2 (v1.26.0[60]);. Gene expression was modeled with cell type (DPPIV⁺/DPPIV⁻ FAPs) and source of cells (animal) specified in the design formula. Wald test was used for differential gene expression and all *P*-values were adjusted by Benjamini-Hochberg correction. The magnitude of expression shown in heatmaps represents relative expression per where regularized log-transformed counts were scaled into *Z*-scores across samples.

## Single-cell RNA sequencing data analysis

Raw barcode-count matrix of quiescent mesenchymal progenitors was downloaded from the Gene Expression Omnibus (accession number: GSE110037). Analysis of sequencing data was performed in R (v4.1.2) using Seurat (v4.3.0[61]);. We isolated for single viable cells by filtering for barcodes based on common quality control metrics, including the number of genes expressed (nFeature_RNA > 200), number of unique molecular identifiers (UMIs <7500), and proportion of mitochondrial transcripts (percent.mt <5%). Normalization was performed using SCTransform (v2) with percent.mt and number of UMIs regressed out as potential confounding variables. This was followed by principal component analysis (PCA), Louvain clustering, and dimensionality reduction via uniform manifold approximation projection (UMAP). Clusters were classified into distinct cell types based on established gene signatures: FAPs (*Pdgfra*), tenocytes (*Tnmd*), and mural cells (Mcam). FAP clusters were further segregated into DPPIV^high (red and dark blue clusters) and DPPIV^low (orange and cyan clusters) populations. Comparison between the two populations using Wilcoxon's rank sum test identified *Ly6a*, *Pi16*, *Dpp4*, and *Csf1* as differentially expressed genes (adjusted *p*-val < 0.05). P-values were adjusted by Benjamini-Hochberg correction. Presentation of gene expression by heatmap reflects relative average expression per gene across cell clusters.

## Statistical analysis

All statistical analysis was done using GraphPad Prism v9. Parametric tests such as *t*-tests and ANOVA were used for statistical comparison. To compare two groups, initially *F*-test was used to compare variances. If there was no difference in variances, statistical comparison was performed using unpaired *t*-test. If the *F*-test showed significantly different variances, unpaired *t*-test with Welch's correction was performed. To compare multiple groups, we used one-way ANOVA. Initially, Brown-Forsythe was used to compare standard deviations. If the test showed no significant difference between standard deviations, ordinary one-way ANOVA followed by a proper multiple comparison test was performed. Throughout the manuscript *p*-value < 0.05 was considered significant. Graphs were generated using Graphpad Prism v9, R version 3.6.2, Adobe Illustrator v. v27.7, or Adobe Photoshop v24.6.0. The figures display the mean ± standard error of the mean.

## Reporting summary

Further information on research design is available in the Nature Portfolio Reporting Summary linked to this article.

## Data availability
The raw RNAseq data have been deposited in the Gene Expression Omnibus (GEO) database under accession code GSE241832. Source data are provided with this paper.

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

## Acknowledgements

We wish to thank the Biomedical Research Centre animal facility, genotyping facility, sequencing facility, AbLab and core staff, as well as the University of British Columbia flow cytometry facility staff for their technical assistance. This work was supported by a grant from the Canadian Institute for Health Research (CIHR) (CIHR-FDN-159908) to F.M.V.R., a grant from US National Institutes of Health (AG045040) and Welch Foundation Grant (AQ-1507) to J.X.J., a CIHR Vanier Canada Graduate Scholarship and a Four-Year Doctoral Fellowship (4YF) by the University of British Columbia to F.B., 4YF from The University of British Columbia and the Dennis Washington Leadership Graduate Scholarship from the Dennis and Phyllis Washington Foundation to L.W.T., and a summer studentship award by UBC Centre for Blood Research/School of Biomedical Engineering to R.L.

## Author contributions

F.B. designed, directed, and carried out experiments; analyzed data; prepared figures; and wrote the manuscript. N.K., R.L, C.W.C., R.C., and T.H. carried out experiments, and analyzed data. L.W.T analyzed data and assisted with the figures. M.R. carried out experiments, analyzed data, and assisted with the figures. E.G., and C.-K.C carried out experiments. J.X.J provided the Csf1 flox/flox mice and edited the manuscript. F.M.V.R. designed experiments, directed the project, and edited the manuscript.

## Competing interests

The authors declare no competing interests.
