## [Peer Review File · Nature Communications]

F 9J =9K 9F `7CA A 9BHG

Reviewer #1 (expert in muscle progenitor cell fate determination, skeletal muscle differentiation and regeneration):

In this manuscript, Babaeijandaghi et al. aimed at characterizing the cellular and molecular composition of the niche that controls the number and the activity of skeletal muscle resident macrophages (RMs) in homeostatic conditions.

To this end, they first look for the local cellular source of colony-stimulating factor 1 (CSF1), the main factor required for the development and maintenance of RMs. Using murine transgenic lines that deplete Fibro-Adipogenic Progenitors (FAPs) (Pdgfr α CreERT2*DTA) or conditionally delete FAP-derived CSF1 (Pdgfr α CreERT2*Csf1flox/null), together with parabiosis techniques, the authors accumulated evidence in favor of FAPs being the main source of CSF1 and the main cell population able to regulate RMs' self-renewal and homeostatic number, locally. Further, based on evidence of local proximity to RMs they point to a specific subpopulation of FAPs, expressing DPPIV, as the specific nurturing cells.

Overall, the paper is clear, well-written and the results are of significance in the field, as they expand the current knowledge on the possible factors involved in the regulation of skeletal muscle RMs offering novel opportunities to manipulate RMs in situ in health and disease conditions.

However, in the current form the paper lacks some important controls and more evidence must be gathered to support authors' conclusions. Specifically:

- Figure 1: authors should show they efficiently succeed to deplete Csf1 specifically in FAPs (in the Pdgfr α CreERT2*Csf1flox/null mice), by either assessing its levels in different populations by qRT-PCR, as they did in Fig.1b; or at least validating it at whole muscle level by ELISA, as they did in fig 1d.
- Figure 1b: since the authors produced single-cell RNAseq data set in skeletal muscle (Scott et al., 2019), this result could have been supported by scRNA-seq evidence of specific enrichment of Csf1 in FAPs, as compared to other muscle-resident populations.

- Fig2c: at which time point is this analysis done? The legend says it is done "at different time points" but the graph shows Csf1 mRNA levels at only one timing. It is important to show how Csf1 levels change in FAPs along the treatment. As, if the model is correct, I would expect to see a peak of Csf1 expression during CSFR1 inhibition and a return to steady state after withdrawal.

Also, does the number of FAPs change upon CSFR1 inhibition? It is important to show this (or to mention it if data in such direction are already available in the literature) to draw the conclusion that it is the level of produced CSF1 that matters and not a different amount of FAPs per se that modulates this response.

- Fig2d: Based on this evidence, the authors say that “CSF1 from FAPs was required for the efficient replenishment of both populations of RMs following their pharmacological depletion in CSF1flox mice”. But the experiment as designed (as far as I understood) does not allow to draw this conclusion. Indeed, FAPs-specific CSF1 depletion induces reduction in the TIM4+ and TIM4- populations regardless of CSF1R inhibition/withdrawal, as also shown in Fig1f.

I understand that they want to emphasize that CSFR1inhibition does not rescue RMs number without FAPs-derived CSF1, but as a better control they should compare Ctrl and Pdgfr-CreERT2/Csf1Flox mice untreated vs those treated with the CSFR1 inhibitor. On the other hand, I also think that to claim that FAPs-derived CSF1 is important to restore depleted resident macrophages, authors should compare Ctrl and Pdgfr-CreERT2/Csf1Flox mice by repeating the same analyses shown in Fig.2a and 2c. They could also validate this in mice depleted of FAPs (Pdgfr α CreERT2 \times DTA mice) vs controls.

- Fig.3a: along the same line of the comments above, is this overshoot (TIM4-) or return to steady-state (TIM4+) levels blunted in the absence of FAPs or in mice with FAPs-specific Csf1 depletion?

- Fig S2a: the conclusion is based on an experiment with n=1, the authors should increase the sample size to draw any conclusion. Also, it is not clear from which mice they isolated GFP+/TIM4- RMs, they should clarify it.

- Figure 3c. Is this distance affected in muscles in which Csf1 is specifically depleted in FAPs? In other words, is the production of Csf1 by FAPs that sustains niche proximity between FAPs and TIM4+ RMs?

Also, as a general comment, the authors based their conclusion on FAPs representing the nurturing niche for self-renewing RMs also in light of the evidence of their physical proximity, as compared to blood-derived RMs. However, is this proximity specific for FAPs? The authors should validate this conclusion by comparing the distance between LYVE1+ and LYVE- RMs and another muscle-resident cell population (i.e. MuSCs) as a control. Also, the authors should verify whether the majority of CD68+ LYVE1+ RMs is in close proximity to FAPs; which is the percentage of CD68+ LYVE1+ RMs located far from FAPs as compared to those closer to them?

- Fig.3d. I understand that the gap is an artifact of tissue preparation, but don't they have a better image to show? I would be surprised if not..

- Fig-S3a. The shown heatmap is confusing and it is hard to appreciate that “Dpp4 was highly expressed in Csf1-expressing FAPs”. I think a clearer way to show this would be a violin plot for Dpp4 between Csf1+ vs Csf1- FAPs. Also, it is not well justified or explained why the authors choose Dpp4 for further analysis, was it the highest in term of expression?

-Fig 4c. While I see the importance to highlight the enrichment of these genes and categories in DPPIV+ FAPs as compared to DPPIV-, this piece of evidence appears rather incomplete and it leaves the readers wonder what it might be the specific role of such molecules, in the model proposed by the authors. I think they should make a bigger effort to add some functional validation of at least some, or also just one, of these factors in mediating FAPs nurturing function towards SRRMs. Also, are these categories among the top enriched in DPPIV+ FAPs?

- Material and methods section lacks any description or reference to the procedure of CSF1R inhibition. I know it has been already performed in a previous paper by the group, but it would be appropriate also a brief description here for the sake of clarity.

Reviewer #2 (expert in macrophage and CSF1R):

This manuscript reports that FAPs are the main source of CSF1 in skeletal muscle and comprise a niche for tim4+ self-renewing resident macrophages, especially DPP4+ FAPS. The expression of CSF1 in DPP4+ vs DPP4- FAPs is shown in re-analysed public scRNA-Seq data, but not in the authors' own populations. Differential replenishment of Tim4+/- populations from circulating cells is also shown. The microscopy data localising the cells in the 'niche' are not overly compelling. Additional data and clarification are required to interpret the data presented.

Specific comments.

- In my view, the combined results and discussion format obscures some of the gaps in this study. There is also quite a bit of introductory material in the results.

Fig s1

- The gating strategy leading to myeloid cells depicted in 1a should be shown – in fig 1A, only 3% of muscle cells are CD45+, what % of these are actually myeloid cells?

Fig 1

- I found it strange the authors chose to measure CSF1 expression in bone marrow in PDGFR-Cre x CSF1 flox model (Fig s1) to conclude that only CSF1 from PDGFR+ FAPs was affected. The impact on circulating CSF1 should be measured. The methods refer to DNA analysis of efficacy of CSF1 deletion in this model but the data are not shown.

- In panels F/G – why did it take 2-3 months to deplete macrophages in the pdgfr-Cre/csf1-flox model, but only 2 weeks in the pdgfr/DTA model depleted? The Tim4-ve population was depleted when FAPs

were depleted, but not significantly when CSF1 was deleted from FAPS (E v F) – could the different timepoint have something to do with this? Why were tim4- depleted in the latter but not the former?

-

Fig 2

- There is no information in the results or methods regarding CSF1R inhibition.
- The extent of macrophage depletion/recovery at the indicated timepoints after withdrawal (3, 5, 7) should be shown. Figure 3A shows depletion and recovery at 7-14 days. CSF1 expression in whole muscle apparently peaks at day 3 and returns to baseline by day 7, yet the resident Tim4+ macrophages are not fully recovered out to 14 days – can the authors comment/discuss?
- Pg/mg protein is an unusual way to present serum csf1 concentration
- What day post-withdrawl does panel c represent?

Fig S2

- No details of intramuscular transplant method are given, nor the identify of the GFP reporter used.
- S2 should show full gating strategy ie GFP positivity of the cells in the recipient – how was the positive gate set?
- No quantification of chimerism in the intramuscular transplant experiment, is this data based on 1 replicate?

-

Fig 3

- I appreciate that IHC on muscle is challenging, however microscopy data are not at all convincing, with only specks of staining. How many cells were actually counted and how many sections in the 6 mice – assuming the distance is an average? How many EDU+ cells were actually present? Why do lyve1+ vessels appear in d but not c?

Fig s3

- The gene enrichment in CSF1+/- FAPs, including Dpp4 is not very convincing, nor is qPCR of csf1 from dpp4+ and dpp4- FAP from n=2 in panel d. The methods mention statistical analysis of differentially expressed genes in this data set – what was the significance level set for A? are these the only differentially expressed genes between the 2 populations?
- How many experiments are panel c and b representative of – n=1

-

- Fig 4

- Similar concerns with microscopy to Fig 3 – how many cells were counted?

- RNA Seq should compare D0 and D3 if the interpretation is that these factors are upregulated when macrophages are depleted, they may simply be expressed at steady state. What does the colour bar represent – ie how large is this change? Why were these factors selected for presentation rather than a complete analysis of the experiment? Was csf1 itself detected in this analysis? Methods say deseq was used but no indication of significance of these factors is given – can this analysis be done on n=2?

-

- lines 293-296 = repeated section from earlier

Methods

- Refer to tie2-cre model which is not reported in the study

- Line 375 refers to DNA analysis of the efficiency of cre-lox deletion of csf1, this is not shown in the results.

- Sequencing data should be made available in a public database.

- No methods for csf1r inhibition are given.

- No methods for intramuscular transplantation.

Reviewer #3 (expert in skeletal muscle immunology, macrophages):

Here Babaeijandaghi et al. investigate the resident macrophage niche in skeletal muscles. This manuscript is complementary to the same groups recent report (Science Translational Medicine 2022) showing that CSF1r inhibitors can deplete resident muscle macrophages affecting muscle fiber metabolic programming. This new manuscript contains a number of noteworthy findings: 1) muscle mesenchymal cells (FAPs) are the primary sources of CSF1 in skeletal muscles; 2) depletion of CSF1 from FAPs [Flox-Csf1; Pdgfra-CreERT2 mouse model] depletes Tim4⁻ monocyte derived macrophages and Tim4⁺ self-renewing macrophages; 3) CSF1 concentration in muscle adjusts relative to concentration of macrophages; 4) Tim4⁺ resident self-renewing macrophages can be replaced by monocyte-derived circulating macrophages; 5) Tim4⁺ resident macrophages reside in close proximity to DPPIV⁺ FAPs suggestive of a supportive role for DPPIV⁺ FAPs in maintaining resident macrophages through CSF1 production. These findings are of great interest to the muscle biology field. Macrophages are surprisingly prominent cells even in steady state muscle that can increase 100-fold with injury or disease. Although our understanding of the role of monocyte derived macrophages in regulating muscle repair and regeneration has greatly improved far less is known about resident muscle macrophages in maintaining tissue health. This experimentation identifies a new regulatory role for FAPs in maintaining resident muscle macrophages.

Major Comments:

- The data that local changes in FAPs or CSF1 control resident macrophages are not convincing. It should be determined whether hematopoiesis (or myelopoiesis) is affected in each mouse model (Flox-Csf1; Pdgfra-CreERT2, PDGFRa-DTA mice) and CSF1 inhibition and withdrawal experiments. This is particularly important given the authors' data showing that circulating cells can replenish muscle RMs in combination with the known functions of CSF1 and to a lesser extent PDGFRa on differentiation of HSCs into macrophages.
- No data demonstrating effective recombination and downregulation of Csf1 in Flox-Csf1; Pdgfra-CreERT2 mice.
- Why inconsistent experimental timecourses? Initial experimentation showed that in vivo depletion of PDGFRa+ cells resulted in great reductions of TIM4- and TIM4+ macrophages within 15-days of tamoxifen cessation. In the ensuing Flox-Csf1; Pdgfra-CreERT2 mouse experiments resident macrophages were not measured until 2-3 months after tamoxifen cessation. Why the difference in experimental timeline? Given the responsiveness of resident macrophages (RMs) to CSF1R inhibitor and PDGFRa depletion a more rapid response would be expected with conditional CSF1 mutants.

Minor Comments:

- High variability in data sampling. The number of muscle resident macrophages quantified from experiment to experiment is highly variable with control samples fluctuating by more than an order of magnitude from experiment to experiment (Fig. 1e vs 1g).
- Data validating the specificity of CSF1 ELISA for tissue based detection and quantification of CSF1.

RESPONSE TO REVIEWERS' COMMENTS

Dear reviewers,

First, we would like to thank you all for the invaluable feedback. We are delighted to see the reviewers' recognition of the novelty and significance of our efforts in characterizing the niche of self-renewing resident macrophages. The manuscript is now significantly revised according to your comments and concerns. Main changes in the text are labeled in blue. Please find below the summary of the main changes being made to the manuscript followed by our detailed response to the concerns:

- 1) We have evaluated the efficiency of *Csf1* deletion in *Pdgfra^{CreERT2} × Csf1^{flox/null}* mice.*
- 2) We have now quantified the distance between each sub-population of resident macrophages and endothelial cells for control purposes.*
- 3) We additionally studied the expression of *Csf1* across various muscle-resident cell types using publicly available single-cell RNAseq datasets.*
- 4) We have further evaluated the effect of CSF1R inhibition on the number of FAPs.*

Reviewer #1 (expert in muscle progenitor cell fate determination, skeletal muscle differentiation and regeneration):

In this manuscript, Babaeijandaghi et al. aimed at characterizing the cellular and molecular composition of the niche that controls the number and the activity of skeletal muscle resident macrophages (RMs) in homeostatic conditions.

To this end, they first look for the local cellular source of colony-stimulating factor 1 (CSF1), the main factor required for the development and maintenance of RMs. Using murine transgenic lines that deplete Fibro-Adipogenic Progenitors (FAPs) (*Pdgfra^{CreERT2}*DTA*) or conditionally delete FAP-derived CSF1 (*Pdgfra^{CreERT2}*Csf1^{flox/null}*), together with parabiosis techniques, the authors accumulated evidence in favor of FAPs being the main source of CSF1 and the main cell population able to regulate RMs' self-renewal and homeostatic number, locally. Further, based on evidence of local proximity to RMs they point to a specific subpopulation of FAPs, expressing DPPIV, as the specific nurturing cells. Overall, the paper is clear, well-written and the results are of significance in the field, as they expand the current knowledge on the possible factors involved in the regulation of skeletal muscle RMs offering novel opportunities to manipulate RMs in situ in health and disease conditions.

However, in the current form the paper lacks some important controls and more evidence must be gathered to support authors' conclusions. Specifically:

- Figure 1: authors should show they efficiently succeed to deplete *Csf1* specifically in FAPs (in the *Pdgfra^{CreERT2}*Csf1^{flox/null}* mice), by either assessing its levels in different populations by qRT-PCR, as they did in Fig. 1b; or at least validating it at whole muscle level by ELISA, as they did in fig 1d.

*We would like to thank the reviewer for suggesting the assessment of *Csf1* depletion under the *Pdgfa-Cre* driver. Successful deletion of *Csf1* in FAPs assessed by droplet digital PCR has been added to the revised manuscript (fig. S1d).*

- Figure 1b: since the authors produced single-cell RNAseq data set in skeletal muscle (Scott et al., 2019), this result could have been supported by scRNA-seq evidence of specific enrichment of *Csf1* in FAPs, as compared to other muscle-resident populations.

The data set from Scott et al. predominantly includes Hic-1⁺ cells, while excluding other resident cell types such as immune or endothelial cells. We have now analyzed single-cell RNAseq data from two other publicly available data sets that encompass the majority of muscle resident cell types. The results have been added to the revised manuscript (fig. S1b-c).

- Fig2c: at which time point is this analysis done? The legend says it is done “at different time points” but the graph shows *Csf1* mRNA levels at only one timing. It is important to show how *Csf1* levels change in FAPs along the treatment. As, if the model is correct, I would expect to see a peak of *Csf1* expression during CSFR1 inhibition and a return to steady state after withdrawal.

*We apologize for a lack of clarification regarding the time point of the analysis. Initially, we showed that *Csf1* expression in skeletal muscle at steady state is confined to FAPs among the cell types we investigated (Fig. 1b). The inquiry into whether FAPs still exclusively express *Csf1* or if the other cell types acquire *Csf1* expression following CSFR1 inhibition was assessed in this experiment. While there is typically a correlation between mRNA and protein levels, a study by Woo, HH., et al. Mol Cancer (2012) showed small correlation between *Csf1* expression and its protein level (Fig. 5 in the paper). This lack of correlation between *Csf1* mRNA and protein levels has also been suggested by others (Wu Z, et al., Regulation and function of macrophage colony-stimulating factor (CSF1) in the chicken immune system. Dev Comp Immunol. 2020). Therefore, we did not extensively investigate potential changes in the level of *Csf1* mRNA in FAPs following treatment. This has been further discussed in the manuscript. As per the request, we have updated the figure to indicate the corresponding time point for each value.*

Also, does the number of FAPs change upon CSFR1 inhibition? It is important to show this (or to mention it if data in such direction are already available in the literature) to draw the conclusion that it is the level of produced CSF1 that matters and not a different amount of FAPs per se that modulates this response.

We thank the reviewer for bringing this point to our attention. As per the request, the effect of CSFR1 inhibition on the number of FAPs has been assessed. We did not detect any significant changes following CSFR1 inhibition (fig. S2)

- Fig2d: Based on this evidence, the authors say that “CSF1 from FAPs was required for the efficient replenishment of both populations of RMs following their pharmacological depletion in CSF1flox mice”. But the experiment as designed (as far as I understood) does not allow to draw this conclusion. Indeed, FAPs-specific CSF1 depletion induces reduction in the TIM4⁺ and TIM4⁻ populations regardless of CSFR1 inhibition/withdrawal, as also shown in Fig1f. I understand that they want to emphasize that CSFR1 inhibition does not rescue RMs number without FAPs-derived CSF1, but as a better control they should compare Ctrl and Pdgfr-CreERT2/*Csf1*Flox mice untreated vs those treated with the CSFR1 inhibitor. On the other hand, I also think that to claim that FAPs-derived CSF1 is important to restore depleted

resident macrophages, authors should compare Ctrl and Pdgfr-CreERT2/Csf1Flox mice by repeating the same analyses shown in Fig.2a and 2c. They could also validate this in mice depleted of FAPs (PdgfrCreERT2 × DTA mice) vs controls.

Thank you to the reviewer for raising this important point. We would like to clarify that the experiments recommended by the reviewer were indeed conducted and are presented in the figures (Fig. 1f for control vs. Csf1^{flox} without CSF1R inhibition, and Fig. 2d for control vs. Csf1^{flox} following CSF1R inhibition). However, we acknowledge the reviewer's valid concern regarding the interpretation of the significance of FAPs in repopulating RMs following their pharmacological depletion. The reduction in the number of RMs after CSF1 depletion in FAPs at steady state could potentially complicate drawing conclusions solely from comparing control mice with treated CSF1^{flox} mice post-CSF1R inhibition. In the revised manuscript, we have expanded the discussion around the data presented in these two figures. Given the observed variability in the kinetics of RMs repopulation after CSF1R inhibition and withdrawal, coupled with the limited recombination efficiency of the Csf1^{flox} system, we have exercised caution in making definitive conclusions. It is worth mentioning that although we couldn't arrive at definitive conclusions concerning the involvement of FAPs in the repopulation of RMs following their pharmacological depletion (Fig. 2), our data strongly supports the notion of DPPIV⁺ FAPs serving as the niche for SRRMs under steady-state conditions (Figures 1, 3, and 4).

- Fig.3a: along the same line of the comments above, is this overshoot (TIM4-) or return to steady-state (TIM4+) levels blunted in the absence of FAPs or in mice with FAPs-specific Csf1 depletion?

Yes, as depicted in Fig. 1f, a decreased number of RMs is evident upon targeted depletion of CSF1 from FAPs in the steady-state muscle. This reduction becomes more pronounced after CSF1R inh/wd (Fig. 2d).

- Fig S2a: the conclusion is based on an experiment with n=1, the authors should increase the sample size to draw any conclusion. Also, it is not clear from which mice they isolated GFP+/TIM4- RMs, they should clarify it.

Please take into account that this preliminary experiment was conducted solely to investigate whether, TIM4⁻ macrophages have the capability to transition into TIM4⁺ macrophages. Following the initial transplantation analysis, which indicated their potential, we conducted parabiosis experiments (n = 6, with 3 pairs per time point) to corroborate this discovery under more physiological conditions. This approach also allowed us to evaluate the degree of contribution from bloodborne cells to the TIM4⁺ cell population following CSF1R inhibition and subsequent withdrawal. (Fig. 3b). We acknowledge the reviewer's suggestion that increasing the sample size would be ideal. In fact, we performed multiple repetitions of this experiment. However, due to the poor engraftment of macrophages following intramuscular injection, the number of successfully engrafted macrophages was extremely low in most of these instances, making it challenging to conduct a thorough analysis. These limitations have been mentioned in the revised manuscript. Supplementary Figure 3a and its legend have been revised to offer additional details. Furthermore, the FACS plots have been updated from contour plots to density plots. This modification aims to more effectively illustrate the limited number of GFP⁺ macrophages that were retrieved post-transplantation.

- Figure 3c. Is this distance affected in muscles in which *Csf1* is specifically depleted in FAPs? In other words, is the production of *Csf1* by FAPs that sustains niche proximity between FAPs and TIM4⁺ RMs?

CSF1 plays a crucial role in the survival of RMs. In cases where Csf1 is specifically depleted in FAPs within muscle, RMs fail to survive. The question of whether the distance between FAPs and RMs is regulated by the gradient of CSF1 concentration is indeed intriguing, yet addressing it in an in vivo context presents significant challenges.

Also, as a general comment, the authors based their conclusion on FAPs representing the nurturing niche for self-renewing RMs also in light of the evidence of their physical proximity, as compared to blood-derived RMs. However, is this proximity specific for FAPs? The authors should validate this conclusion by comparing the distance between LYVE1⁺ and LYVE⁻ RMs and another muscle-resident cell population (i.e. MuSCs) as a control. Also, the authors should verify whether the majority of CD68⁺ LYVE1⁺ RMs is in close proximity to FAPs; which is the percentage of CD68⁺ LYVE1⁺ RMs located far from FAPs as compared to those closer to them?

To determine the proportion of CD68⁺ LYVE1⁺ RMs situated in close proximity to FAPs, we initially need to establish a distance threshold, which could introduce subjectivity. In order to adopt a more objective approach, we compared the distances between each sub-population of RMs and FAPs. We completely agree with the reviewer that comparing the distance between each sub-population of RMs and another muscle resident cell population would contribute to validating our findings. However, MuSCs are not suitable candidates due to their significantly lower abundance compared to RMs, and notably, their location beneath the basal lamina. As a result, we now have quantified the distance between each sub-population of RMs and endothelial cells, which constitute another abundant population of muscle resident stromal cells (fig. S3b).

- Fig.3d. I understand that the gap is an artifact of tissue preparation, but don't they have a better image to show? I would be surprised if not.

*Unfortunately, the number of EdU⁺LYVE1⁺ RMs in each TA section is limited due to their low proliferation rate at steady state. The selection of this particular spot for presentation was based on that the figure displays **two** LYVE1⁺ EdU⁺ cells situated in proximity to FAPs, and more importantly one of the cells exhibits a dividing nucleus, which has now been illustrated in greater detail in the inset. The sentence concerning the artifact of tissue preparation has also been removed from the revised manuscript as it was unnecessary and to avoid potential confusion for the readers.*

- Fig-S3a. The shown heatmap is confusing and it is hard to appreciate that "Dpp4 was highly expressed in *Csf1*-expressing FAPs". I think a clearer way to show this would be a violin plot for Dpp4 between *Csf1*⁺ vs *Csf1*⁻ FAPs. Also, it is not well justified or explained why the authors choose Dpp4 for further analysis, was it the highest in term of expression?

*We concur with the reviewer and have updated the heatmap to show the average gene expression per cell subset rather than expression value for each cell. Higher expression of *Csf1* by Dpp4⁺ FAPs is now evident (fig. S4a). As shown in the heatmap, Dpp4 was among*

the genes that were highly expressed in Csf1⁺ FAPs. Multiple studies have distinguished skeletal muscle FAPs into Dpp4⁺ and Dpp4⁻ subsets, both of which are functionally distinct (Oprescu SN, et al., iScience. 2020; Scott RW, et al., Cell Stem Cell. 2019). Therefore, Dpp4 was chosen for further analysis. This has been mentioned in the manuscript.

-Fig 4c. While I see the importance to highlight the enrichment of these genes and categories in DPPIV⁺ FAPs as compared to DPPIV⁻, this piece of evidence appears rather incomplete and it leaves the readers wonder what it might be the specific role of such molecules, in the model proposed by the authors. I think they should make a bigger effort to add some functional validation of at least some, or also just one, of these factors in mediating FAPs nurturing function towards SRRMs.

We appreciate the reviewer's suggestion. The primary objective of this study was to elucidate the niche of muscle SRRMs. We selected to highlight differences in expression of BMP, WNT and Semaphorin genes as they have been reported by others to moderate the activity, function, and fate of myelomonocytic cells (Capucha T, et al., J Exp Med. 2018; Malsin ES, et al., Front Immunol. 2019; Kanth SM, et al., Front Immunol. 2021). While the interaction between SRRMs and their niche is indeed a captivating area of inquiry, its complexity extends beyond the confines of this study. We aspire for our findings and preliminary sequencing data to serve as a foundation for further exploration of this intriguing subject matter.

Also, are these categories among the top enriched in DPPIV⁺ FAPs?

Yes. Those categories are among the top enriched in DPPIV⁺ FAPs. This has been mentioned in the figure legend.

- Material and methods section lacks any description or reference to the procedure of CSF1R inhibition. I know it has been already performed in a previous paper by the group, but it would be appropriate also a brief description here for the sake of clarity.

We thank the reviewer for bringing this point to our attention. A brief description of the procedure has been added to the revised manuscript.

Reviewer #2 (expert in macrophage and CSF1R):

This manuscript reports that FAPs are the main source of CSF1 in skeletal muscle and comprise a niche for tim4⁺ self-renewing resident macrophages, especially DPP4⁺ FAPS. The expression of CSF1 in DPP4⁺ vs DPP4⁻ FAPs is shown in re-analysed public scRNA-Seq data, but not in the authors' own populations. Differential replenishment of Tim4⁺/⁻ populations from circulating cells is also shown. The microscopy data localising the cells in the 'niche' are not overly compelling. Additional data and clarification are required to interpret the data presented.

Specific comments.

- In my view, the combined results and discussion format obscures some of the gaps in this study. There is also quite a bit of introductory material in the results.

Fig s1

- The gating strategy leading to myeloid cells depicted in 1a should be shown – in fig 1A, only 3% of muscle cells are CD45+, what % of these are actually myeloid cells?

The flow cytometry gating strategy leading to myeloid cells has now been updated in the revised manuscript. Based on the data from our two previous studies (Babaeijandaghi F., et al., Science Translational Medicine, 2022; Babaeijandaghi F., et al., PNAS, 2022) and as is shown in fig. S4c, very little B cells and T cells reside in skeletal muscle at steady state and the majority (80-90%) of CD45⁺ cells are myeloid cells.

Fig 1

- I found it strange the authors chose to measure CSF1 expression in bone marrow in PDGFR-Cre x CSF1 flox model (Fig s1) to conclude that only CSF1 from PDGFR⁺ FAPs was affected. The impact on circulating CSF1 should be measured. The methods refer to DNA analysis of efficacy of CSF1 deletion in this model but the data are not shown.

We measured CSF1 in the bone marrow in the PDGFR α ^{CreERT2} × DTA mouse model, as PDGFR α ⁺ cells also reside in the bone marrow. Moreover, we also did not see any change in blood CSF1 level following pharmacological depletion of RMs (Fig. 2b). We apologize for the oversight. DNA analysis of Csf1 deletion has now been added to the revised manuscript (fig. S1d)

- In panels F/G – why did it take 2-3 months to deplete macrophages in the pdgfr-Cre/csf1-flox model, but only 2 weeks in the pdgfr/DTA model depleted? The Tim4-ve population was depleted when FAPs were depleted, but not significantly when CSF1 was deleted from FAPS (E v F) – could the different timepoint have something to do with this? Why were tim4-depleted in the latter but not the former?

We appreciate the reviewer for bringing up this point. This might be attributed, in part, to the relatively lower recombination efficiency of the CSF1^{flox} system as well as using mice with a null allele of Csf1 as controls. It may also indicate that FAPs supply additional physical and/or molecular signals capable of temporarily compensating for the lack of CSF1, and thereby supporting the survival of RMs. Furthermore, using mice with a null allele of Csf1 as controls could also potentially explain the relatively modest reduction in the number of RMs observed in Fig. 1f. As is depicted in the Fig. 1g, more pronounced reduction in the number of RMs is observed when a comparison was made to the WT parabiotic partner. This has been further discussed in the revised manuscript.

Fig 2

- There is no information in the results or methods regarding CSF1R inhibition.

Thanks to the reviewer for bringing this point to our attention. The methods of CSF1R inhibition have been added to the revised manuscript.

- The extent of macrophage depletion/recovery at the indicated timepoints after withdrawal (3, 5, 7) should be shown. Figure 3A shows depletion and recovery at 7-14 days. CSF1 expression in whole muscle apparently peaks at day 3 and returns to baseline by day 7, yet the resident $TIM4^+$ macrophages are not fully recovered out to 14 days – can the authors comment/discuss?

According to our previous study (Babaeijandaghi F., et al., *Science Translational Medicine*, 2022), SRRMs regain their numbers within 7 days after CSF1R withdrawal:

However, in this study, the number of SRRMs was slightly lower than the baseline after 7-14 days of drug withdrawal:

We attribute this variation primarily to biological variability, which is evident in the extensive number of WT (Control) mice we included in our previous dataset (the first figure above). Despite this biological variability, we consistently observed in both trials that after drug withdrawal, the count of blood-derived $TIM4^-$ cells significantly exceeded their pre-treatment levels, whereas the number of $TIM4^+$ SRRMs remained near pre-treatment levels.

- Pg/mg protein is an unusual way to present serum csf1 concentration.

Serum concentration of CSF1 is now reported as mg/mL.

- What day post-withdrawl does panel c represent?

As per the reviewer's request, we have updated the figure to indicate the corresponding time point for each value.

Fig S2

- No details of intramuscular transplant method are given, nor the identify of the GFP reporter used.

Please refer to our answer below.

- S2 should show full gating strategy ie GFP positivity of the cells in the recipient – how was the positive gate set?

Please refer to our answer below.

- No quantification of chimerism in the intramuscular transplant experiment, is this data based on 1 replicate?

As mentioned in the response to a similar comment from reviewer one, this preliminary experiment was conducted solely to investigate whether, TIM4⁻ macrophages have the capability to transition into TIM4⁺ RMs (n =1). In fact, we performed multiple repetitions of this experiment. However, it's important to highlight that due to the poor engraftment of macrophages following intramuscular injection, the number of successfully engrafted macrophages was extremely low in most of these instances, making it challenging to conduct a thorough analysis. These limitations have been mentioned in the revised manuscript. Following the initial transplantation analysis, which indicated their potential, we conducted parabiosis experiments (n = 6, with 3 pairs per time point) to corroborate this discovery under more physiological conditions. This approach also allowed us to evaluate the degree of contribution from bloodborne cells to the TIM4⁺ cell population following CSF1R inhibition and subsequent withdrawal (Fig. 3b). The supplementary figure 3a has also been updated to better show our experimental design.

-

Fig 3

- I appreciate that IHC on muscle is challenging, however microscopy data are not at all convincing, with only specks of staining. How many cells were actually counted and how many sections in the 6 mice – assuming the distance is an average? How many EDU⁺ cells were actually present? Why do lyve1⁺ vessels appear in d but not c?

To measure the distance between each subpopulation of resident macrophages and FAPs, we counted cells from both TAs of each mouse (one section each). On average, we manually measured distances of approximately 575 CD68⁺LYVE1⁻ and 450 CD68⁺LYVE1⁺ resident

macrophages from their closest FAPs for each mouse. We strongly believe that the large number of cells assessed per mouse to quantify distance enabled us to make strong conclusions. As the number of EDU⁺ cells were much lower, cells from both TAs of each mouse (3 sections each, 6 sections in total) were counted. On average, we manually measured distances of approximately 175 LYVE1⁺EdU⁻ and 25 EdU⁺ cells from their closest FAPs for each mouse. This information has been added to the figure legend. Regarding the LYVE1⁺ cell in panel d, we want to clarify that it is not a vessel as there is no lumen. Instead, it represents a dividing cell. We apologize if this caused any confusion. To provide better presentation, an inset has been incorporated into the image.

Fig s3

- The gene enrichment in CSF1⁺/⁻ FAPs, including Dpp4 is not very convincing, nor is qPCR of csf1 from dpp4⁺ and dpp4⁻ FAP from n=2 in panel d. The methods mention statistical analysis of differentially expressed genes in this data set – what was the significance level set for A? are these the only differentially expressed genes between the 2 populations?

Thanks to the reviewer for bringing this point to our attention. The heatmap has been updated to show the average gene expression per cell type rather than expression value for each cell. Higher expression of Csf1 by Dpp4⁺ FAPs is now evident (fig. S4a). Genes presented are all differentially expressed (adjusted p-values are below 0.05). However, these are not the only differentially expressed genes. In addition to Csf1 and Dpp4, only key genes identifying different cell populations are presented. The full list of differentially expressed genes is available and can be provided upon request.

- How many experiments are panel c and b representative of – n=1

The existence of a specific subset of FAPs expressing DPP4 has been previously demonstrated by other studies (Oprescu SN, et al., iScience. 2020; Scott RW, et al., Cell Stem Cell. 2019). The primary objective of this experiment, however, was to establish that TIM4⁺ RMs do not express DPPIV. The absence of DPPIV expression on TIM4⁺ RMs would allow us to confidently analyze their spatial relationship with DPPIV⁺ FAPs in muscle histology sections. Our initial screening analysis (Fig. S4b, n = 1) indicated the presence of a subset of CD45⁺ cells expressing DPPIV. Subsequent comprehensive analysis (Fig. S4c, n = 4) revealed that resident macrophages do not express DPPIV, and its expression among CD45⁺ cells is confined to a specific subpopulation of CD11c⁺ cells. We have included the sample size in the figure legend and incorporated mean ± SD values into the figure for clarity.

- Fig 4

- Similar concerns with microscopy to Fig 3 – how many cells were counted?

The figure legend now includes the number of quantified cells. On average, approximately 800 FAPs per mouse were included for quantification. Among them, an average of about 100 FAPs had SRRMs in close vicinity.

- RNA Seq should compare D0 and D3 if the interpretation is that these factors are upregulated when macrophages are depleted, they may simply be expressed at steady state.

The focus of this experiment was not to determine whether these factors are upregulated following CSF1R inhibition. To support our hypothesis that DPPIV⁺ FAPs constitute the niche of SRRMs, our objective was to identify signals that DPPIV⁺ FAPs provide to facilitate replenishment of SRRMs. Therefore, we compared the transcription profile of this cell population to its DPPIV⁻ counterpart at D3 with RNA-sequencing. D3 was selected as it corresponds to the time point when the niche of SRRMs is empty and receptive, presumably providing signals for SRRM replenishment.

What does the colour bar represent – ie how large is this change? Why were these factors selected for presentation rather than a complete analysis of the experiment? Was csf1 itself detected in this analysis? Methods say deseq was used but no indication of significance of these factors is given – can this analysis be done on n=2?

We apologize for the lack of clarification regarding RNAseq analysis. As mentioned by the reviewer, we had used DESeq2 to perform linear modeling and differential gene expression on all genes. Genes presented in Figure 4C are differentially expressed (adjusted p-values are below 0.05) and are selected for presentation given their association with the discussed signaling pathways. The magnitude of expression shown in the heatmap represents relative expression per gene across all samples. This is generated by scaling regularized log-transformed counts into Z-scores. Csf1 is differentially expressed and upregulated in DPPIV⁺ FAPs (log2FC = 0.7, adjusted p-val = 2.65E-12). This is also in agreement with the results of single-cell RNAseq analysis in fig. S4a.

Regarding the number of biological replicates needed for RNAseq analysis, it is accepted to have two replicates when using DESeq2 as there are sufficient degrees of freedom to estimate dispersion. In general, having a low number of biological replicates signifies a reduced “power” to identify true positives in differential gene expression, and therefore yields less significant results. On the contrary, having a higher number of biological replicates correlates with increasing false positives. Nonetheless, these are general guidelines and case-by-case considerations should be taken into account. In our case, even with reduced power, we were able to detect a significant difference between the two groups (1918 differentially expressed genes) and identify patterns that translate into biologically sound interpretations. The lead author of DESeq2 has provided a similar response to previous queries in the following links:

<https://support.bioconductor.org/p/106403/#:~:text=Yes%2C%20as%20long%20as%20there,run%20with%201%20vs%202.>

<https://support.bioconductor.org/p/105382/>

- lines 293-296 = repeated section from earlier

We thank the reviewer for bringing this to our attention. We do apologize for the oversight. It has been corrected in the revised manuscript.

Methods

- Refer to tie2-cre model which is not reported in the study

We thank the reviewer for bringing this point to our attention. To generate Csf1^{null/flox} (Csf1^{flox}) mice, non-tissue-specific null alleles of Csf1 were obtained through a cross

between *Tie2Cre* (The Jackson Laboratory, #008863) with *Csf1^{flox/flox}* mice (a gift of S.A. Werner, University of Texas Health Science Center at San Antoni). This information has been added to the revised manuscript.

- Line 375 refers to DNA analysis of the efficiency of cre-lox deletion of *csf1*, this is not shown in the results.

We do apologize for the oversight. The data has been added to the revised manuscript (fig. S1d)

- Sequencing data should be made available in a public database.

Absolutely. The sequencing data have uploaded onto the Gene Expression Omnibus repository under the accession ID GSE241832.

- No methods for *csf1r* inhibition are given.

Methods for CSF1R inhibition have been added to the revised manuscript.

- No methods for intramuscular transplantation.

Intramuscular transplantation has now been discussed in the body of the manuscript: To assess whether blood-derived, TIM4⁻ macrophages are capable of directly contributing to TIM4⁺ SRRMs, we initially treated C57BL/6J mice with a CSF1R inhibitor to ablate TIM4⁺ SRRMs and empty their niche. Then, we withdrew the inhibition 24 hours before injecting (transplanting) sorted skeletal muscle GFP⁺TIM4⁻ RMs into the Tibialis Anterior muscle. Six days after transplantation, we analyzed whether any of the injected TIM4⁻ cells were able to differentiate into TIM4⁺ cells. Macrophages exhibit poor engraftment following intramuscular injection, and we were able to only detect a small number of engrafted cells out of thousands of cells that were administered. Our preliminary data, however, demonstrated an increase in TIM4 expression within a portion of the recovered GFP⁺TIM4⁻ cells (fig. S3a). This implies the potential for TIM4⁻ cells to differentiate into TIM4⁺ cells under these circumstances. Supplementary Figure 3a and its legend have been revised to offer additional details. Furthermore, the FACS plots have been updated from contour plots to density plots. This modification aims to more effectively illustrate the limited number of GFP⁺ macrophages that were retrieved post-transplantation.

Reviewer #3 (expert in skeletal muscle immunology, macrophages):

Here Babaeijandaghi et al. investigate the resident macrophage niche in skeletal muscles. This manuscript is complementary to the same groups recent report (Science Translational Medicine 2022) showing that CSF1r inhibitors can deplete resident muscle macrophages affecting muscle fiber metabolic programming. This new manuscript contains a number of noteworthy findings: 1) muscle mesenchymal cells (FAPs) are the primary sources of CSF1 in skeletal muscles; 2) depletion of CSF1 from FAPs [Flox-Csf1; Pdgfra-CreERT2 mouse model] depletes Tim4⁻ monocyte derived macrophages and Tim4⁺ self-renewing macrophages; 3) CSF1 concentration in muscle adjusts relative to concentration of macrophages; 4) Tim4⁺ resident self-renewing macrophages can be replaced by monocyte-

derived circulating macrophages; 5) Tim4⁺ resident macrophages reside in close proximity to DPPIV⁺ FAPs suggestive of a supportive role for DPPIV⁺ FAPs in maintaining resident macrophages through CSF1 production. These findings are of great interest to the muscle biology field. Macrophages are surprisingly prominent cells even in steady state muscle that can increase 100-fold with injury or disease. Although our understanding of the role of monocyte derived macrophages in regulating muscle repair and regeneration has greatly improved far less is known about resident muscle macrophages in maintaining tissue health. This experimentation identifies a new regulatory role for FAPs in maintaining resident muscle macrophages.

Major Comments:

- The data that local changes in FAPs or CSF1 control resident macrophages are not convincing. It should be determined whether hematopoiesis (or myelopoiesis) is affected in each mouse model (Flox-Csf1; Pdgfra-CreERT2, PDGFRa-DTA mice) and CSF1 inhibition and withdrawal experiments. This is particularly important given the authors' data showing that circulating cells can replenish muscle RMs in combination with the known functions of CSF1 and to a lesser extent PDGFRa on differentiation of HSCs into macrophages.

Thank you to the reviewer for raising this point. CSF1 depletion would likely have some effect on myelopoiesis (Lei F, et al., Proc Natl Acad Sci (2020)). We also previously showed that the ratio of Ly6C^{Hi}/Ly6C^{Low} blood monocytes changes following CSF1R inhibition (Babaeijandaghi F, et al., Sci Trans Med, 2022, Supp. Fig. 3B). Therefore, to differentiate between the local and systemic effect of CSF1 depletion on muscle resident macrophages, we performed long term parabiosis, a challenging yet essential experiment. According to the points summarized below, our data suggest that CSF1 is produced locally to maintain or restore muscle resident macrophages.

1) *Exposing skeletal muscle lacking CSF1 from FAPs (Pdgfra^{CreERT2} × Csf1^{flox}) to WT blood cells (CD45.1) using long-term parabiosis, did not result in replenishment of resident macrophages, strongly suggesting that CSF1 from local FAPs is required for survival of both populations of RMs (Fig. 1g).*

2) *There was a significant fivefold increase in CSF1 concentration in skeletal muscle following CSF1R inhibition, which returned to baseline levels within 7 days of withdrawing the inhibitor (Fig. 2a). We did not detect any significant changes in CSF1 concentration in the blood following CSF1R inhibition and withdrawal (Fig. 2b), suggesting that the level of CSF1 in skeletal muscle is controlled locally.*

- No data demonstrating effective recombination and downregulation of Csf1 in Flox-Csf1; Pdgfra-CreERT2 mice.

As per the reviewer's request, data demonstrating effective recombination has been added to the revised manuscript (fig. S1d).

Why inconsistent experimental timecourses? Initial experimentation showed that in vivo depletion of PDGFRa⁺ cells resulted in great reductions of TIM4⁻ and TIM4⁺ macrophages within 15-days of tamoxifen cessation. In the ensuing Flox-Csf1; Pdgfra-CreERT2 mouse experiments resident macrophages were not measured until 2-3 months after tamoxifen cessation. Why the difference in experimental timeline? Given the responsiveness of resident

macrophages (RMs) to CSF1R inhibitor and PDGFRa depletion a more rapid response would be expected with conditional CSF1 mutants.

Thank you to the reviewer for raising this important point. Pharmacological inhibition of CSF1R signaling or depletion of FAPs led to a swift reduction in RMs. However, the depletion of FAP-derived CSF1 resulted in a decrease in both TIM4⁺ and TIM4⁻ populations only after a span of 2-3 months of treatment. Notably, no significant changes were observed in the number of both populations within the initial month of CSF1 depletion in FAPs. As mentioned in the response to a similar comment from Reviewer #2, this might be attributed, in part, to the relatively lower recombination efficiency of the CSF1^{fllox} system as well as using mice with a null allele of Csf1 as controls. It may also indicate that FAPs supply additional physical and/or molecular signals capable of temporarily compensating for the lack of CSF1, and thereby supporting the survival of RMs. This has been further discussed in the revised manuscript.

Minor Comments:

- High variability in data sampling. The number of muscle resident macrophages quantified from experiment to experiment is highly variable with control samples fluctuating by more than an order of magnitude from experiment to experiment (Fig. 1e vs 1g).

We thank the reviewer for bringing this point to our attention. Because there is a minor decrease in muscle mass after depleting FAPs, the number of RMs in Fig. 1e is presented per muscle (gastrocnemius) and not per milligram of tissue. The Y-axis title has been corrected accordingly, and this clarification has been included in the legend. We apologize for any confusion caused by the oversight.

- Data validating the specificity of CSF1 ELISA for tissue based detection and quantification of CSF1.

This kit has been widely used by different studies to measure CSF1 in tissue homogenates:

- 1) *Elevated expression of the colony-stimulating factor 1 (CSF1) induces prostatic intraepithelial neoplasia dependent of epithelial-Gp130. Oncogene, 2022*
- 2) *A Macrophage Colony-Stimulating Factor-Producing $\gamma\delta$ T Cell Subset Prevents Malarial Parasitemic Recurrence. Immunity, 2018*
- 3) *Temporal changes in myeloid cells in the cervix during pregnancy and parturition. J. Immunol., 2009*
- 4) *The cell-surface isoform of colony stimulating factor 1 (CSF1) restores but does not completely normalize fecundity in CSF1-deficient mice. Biol. Reprod., 2005*

REVIEWERS' COMMENTS

Reviewer #1 (expert in muscle progenitor cell fate determination, skeletal muscle differentiation and regeneration):

The authors included the experimental controls as requested and properly addressed further concerns.

Reviewer #2 (expert in macrophage and CSF1R):

The authors have previously published on the dynamics of *tim4*^{+/-} skeletal muscle macrophages. most novel finding in this paper is that FAPs, specifically DPPIV⁺ FAPs are a source of CSF1, and specifically form a 'niche' for *Tim4*⁺ resident macrophages. This remains the weakest part of the paper.

The CSF1/flox model requires further explanation. According to the methods it was derived from a cross between *Tie2-Cre* and *Csf1 flox/flox*. This might generate 1 null allele in *Tie2*⁺ cells, presumably this is then crossed with the *PDGFR-Cre-ERT2 x DTA* system? Very difficult to interpret and rationale has not been explained. Whereas changes in CSF1 expression (mRNA or protein) are shown other tissues, CSF1 depletion is shown at the DNA level for the *PDGFRa-Cre* model in the revised manuscript. Why?

The difference in CSF1 expression between DPPIV⁺ and -ve FAPS is marginal (Fig s4D, n=2).

Additional comments:

The flow cytometry gating strategy was provided in the revised manuscript as requested. The gating strategy appears different to the previous STM publication. In the new S1a, CD45 is not used and macrophages are apparently gated as *ly6g*⁺/*siglec*⁺/*cd11b*⁺. In the previous STM paper, 21% of *cd45*⁺ cells were *f480/cd11b*⁺, and these cells were specifically gated as *ly6g*-*siglec*⁻

Why the difference?

Details of CSF1R inhibition were provided in the methods in the revised version, as requested. "PLX5622 hemifumarate (MedChemExpress, # HY-114153A, fig. S2) or PLX73086 (Plexikon Inc., for all other experiments) was used for CSF1R inhibition".

- It is not indicated anywhere which experiment used which compound. What does 'for all other experiments' mean?

- PLX compounds would deplete macrophages in other tissues not just muscle, so why do the authors think there is no change in circulating CSF1? – it increases massively and rapidly upon treatment with anti-csf1r antibody eg afs98

Reviewer #3 (expert in skeletal muscle immunology, macrophages):

In response to my initial review the authors have addressed most of my major criticisms. I question the speculative answer about experimental timeline with regard to inducible CSF1 knockdown. Was this empirically tested? Still, I feel that the results represent a meaningful contribution to the field and feel that my other concerns were sufficiently addressed.

We are delighted to see that Reviewers # 1 and # 3 are satisfied with the current version of the manuscript. Please find below our detailed response to the concerns raised by Reviewer # 2 and # 3:

Reviewer #2 (expert in macrophage and CSF1R):

The authors have previously published on the dynamics of tim4^{+/-} skeletal muscle macrophages. most novel finding in this paper is that FAPs, specifically DPPIV⁺ FAPs are a source of CSF1, and specifically form a 'niche' for Tim4⁺ resident macrophages. This remains the weakest part of the paper.

The CSF1/flox model requires further explanation. According to the methods it was derived from a cross between Tie2-Cre and Csf1 flox/flox. This might generate 1 null allele in Tie2⁺ cells, presumably this is then crossed with the PDGFR-Cre-ERT2 x DTA system? Very difficult to interpret and rationale has not been explained.

We would like to thank the reviewer to bring this into our attention. To generate Csf1^{null/flox} mice, non-tissue-specific null alleles of Csf1 were obtained through a cross between Tie2-Cre with Csf1 flox/flox mice. Tie2-Cre recombinase activity is detected in endothelial tissues, as well as reproductive tissues, leading to the germline deletion of the floxed allele. Subsequently, we selected Csf1null/flox mice and crossed them with PDGFRa-Cre-ERT2 mice to conditionally delete the Csf1 floxed allele in FAPs. This has been added to the revised manuscript.

Whereas changes in CSF1 expression (mRNA or protein) are shown other tissues, CSF1 depletion is shown at the DNA level for the PDGFRa-Cre model in the revised manuscript. Why?

In contrast to mRNA and protein levels, which can vary under different conditions, including sample processing, quantifying the absolute copy number of Csf1 alleles by droplet digital PCR provides a more accurate method for assessing recombination efficiency.

The difference in CSF1 expression between DPPIV⁺ and -ve FAPS is marginal (Fig s4D, n=2).

This data was included solely to demonstrate that CSF1 production is not confined to the niche, as both DPPIV⁻ and DPPIV⁺ FAPs express Csf1. Both bulk sequencing analysis (Fig. S4d) and single-cell RNA sequencing analysis (Fig. S4a) revealed that DPPIV⁺ FAPs exhibit higher levels of Csf1 expression.

Additional comments:

The flow cytometry gating strategy was provided in the revised manuscript as requested. The gating strategy appears different to the previous STM publication. In the new S1a, CD45 is not used and macrophages are apparently gated as ly6g⁺/siglec⁺/cd11b⁺. In the previous STM paper, 21% of cd45⁺ cells were f480/cd11b⁺, and these cells were specifically gated as ly6g-siglec⁻

Why the difference?

We appreciate the reviewer's valuable input. In flow cytometric analysis of skeletal muscle, it is evident that CD11b alone (without CD45) is sufficient for identifying myelomonocytic cells. In current manuscript, we have substituted CD45 with CCR2 for a more comprehensive characterization of these cells. In both studies, we indeed excluded Siglec-F⁺ and Ly6G⁺ cells to specifically identify macrophages. In the panel, you can find the Ly6G⁺ cells, which constitute a small population at the top-middle position (this has been clearly labeled in the revised figure). Under steady-state conditions, Siglec-F⁺ eosinophils are present in very limited numbers in muscle samples, and they have already been excluded based on our initial FSC/SSC gating strategy, as they exhibit higher SSC values. This has been mentioned in the legend and the figure has been updated to eliminate any potential confusion.

Details of CSF1R inhibition were provided in the methods in the revised version, as requested. “PLX5622 hemifumarate (MedChemExpress, # HY-114153A, fig. S2) or PLX73086 (Plexxikon Inc., for all other experiments) was used for CSF1R inhibition”.

- It is not indicated anywhere which experiment used which compound. What does ‘for all other experiments’ mean?

We would like to thank the reviewer to bring this into our attention. PLX5622 hemifumarate was used only in experiments shown fig. S2 and fig. S5. PLX73086 was used for CSF1R inhibition in all other experiments. The manuscript has been updated to include this information.

- PLX compounds would deplete macrophages in other tissues not just muscle, so why do the authors think there is no change in circulating CSF1? – it increases massively and rapidly upon treatment with anti-csf1r antibody eg afs98.

Very recently, Bosch A.J.T. et al. (Diabetologia, 2023) conducted a comprehensive study on the impact of CSF1R inhibition using both a small molecule (PLX5562) and an antibody (AFS98) on various tissue-resident macrophages. Their findings revealed that while PLX5562 was highly effective at depleting approximately 90% of resident macrophages in the colon, peritoneal cavity, and the brain, its effects were considerably less pronounced within the hematopoietic system, such as blood monocytes and spleen macrophages. Moreover, they observed an increased number of Kupffer cells following CSF1R inhibition, and similar outcomes were reported for AFS98. Hence, the impact of CSF1R inhibition appears to be tissue-dependent. In our study, we observed a slight increase in CSF1 concentration in the blood (Fig. 2b), but this increase was negligible when compared to the significant changes we observed in muscle tissue (Fig. 2a).

Reviewer #3 (expert in skeletal muscle immunology, macrophages):

In response to my initial review the authors have addressed most of my major criticisms. I question the speculative answer about experimental timeline with regard to inducible CSF1 knockdown. Was this empirically tested? Still, I feel that the results represent a meaningful contribution to the field and feel that my other concerns were sufficiently addressed.

We are pleased to note that the reviewer is satisfied with the present version of the manuscript. As the reviewer has already guessed, the experimental timeline was indeed empirically tested.